# Cloud droplet size distribution broadening during diffusional growth: ripening amplified by deactivation and reactivation

Fan Yang[1], Pavlos Kollias[1,2], Raymond A. Shaw[3], and Andrew M. Vogelmann[1]

[1]Brookhaven National Laboratory, Upton, New York, USA.
[2]Stony Brook University, Stony Brook, New York, USA.
[3]Michigan Technological University, Houghton, Michigan, USA.

*Correspondence to:* Fan Yang (fanyang@bnl.gov)

**Abstract.** Cloud droplet size distributions (CDSDs), which are related to cloud albedo and rain formation, are usually broader in warm clouds than predicted from adiabatic parcel calculations. We investigate a mechanism for the CDSD broadening using a moving-size-grid cloud parcel model that considers the condensational growth of cloud droplets formed on polydisperse, sub-micrometer aerosols in an adiabatic cloud parcel that undergoes vertical oscillations, such as those due to cloud circulations or turbulence. Results show that the CDSD can be broadened during condensational growth as a result of Ostwald ripening amplified by droplet deactivation and reactivation, which is consistent with early work. The relative roles of the solute effect, curvature effect, deactivation and reactivation on CDSD broadening are investigated. Deactivation of smaller cloud droplets, which is due to the combination of curvature and solute effects in the downdraft region, enhances the growth of larger cloud droplets and thus contributes particles to the larger size end of the CDSD. Droplet reactivation, which occurs in the updraft region, contributes particles to the smaller size end of the CDSD. In addition, we find that growth of the largest cloud droplets strongly depends on the residence time of cloud droplet in the cloud rather than the magnitude of local variability in the supersaturation fluctuation. This is because the environmental saturation ratio is strongly buffered by numerous smaller cloud droplets. Two necessary conditions for this CDSD broadening, which generally occur in the atmosphere, are: (1) droplets form on aerosols of different sizes and (2) the cloud parcel experiences upwards and downwards motions. Therefore we expect that this mechanism for CDSD broadening is possible in real clouds. Our results also suggest it is important to consider both curvature and solute effects before and after cloud droplet activation in a cloud model. The importance of this mechanism compared with other mechanisms on cloud properties should be investigated through in-situ measurements and 3-D dynamic models.

## 1 Introduction

Warm clouds play a crucial role in the water cycle and energy balance on Earth (Boucher et al., 2013). Understanding the whole life cycle of warm clouds, including formation, development and precipitation, is important for better prediction of local weather and global climate. Cloud droplet growth is dominated by diffusion of water vapor at the early stage of cloud development, while collisional growth is considered to be the most important mechanism for drizzle formation and warm cloud precipitation (Pruppacher and Klett, 2010). The concept of a cloud parcel rising adiabatically in the atmosphere has been used

to study cloud microphysical properties for decades. Imagining an initially sub-saturated air parcel rising adiabatically, cloud forms at the lifting condensation level and the growth of cloud droplets due to diffusional growth can be accurately predicted if we know the aerosol chemical composition. Because the growth rate of a cloud droplet is inversely proportional to droplet size, diffusional growth is inefficient when the droplet diameter is larger than 20 $\mu m$. On the other hand, collisional growth is efficient when the droplet diameter is larger than 38 $\mu m$ (Pruppacher and Klett, 2010). Meanwhile, the sizes of the smaller cloud droplets will approach those of the larger droplets and narrow the cloud droplet size distribution (CDSD), which is also unfavorable for collisional growth (Howell, 1949; Mordy, 1959). If only diffusional growth is considered, the CDSD becomes narrower and several tens of minutes even up to hours will be needed for a cloud droplet to reach efficient-collision size in an ascending cloud parcel. However, the CDSD in a real cloud is usually wider than predicted by an adiabatic cloud parcel model and drizzle-size cloud droplets are frequently observed in warm clouds (e.g., Laird et al., 2000; Glienke et al., 2017; Siebert and Shaw, 2017).

The broadening of the CDSD has a strong effect on precipitation and radiation. A broader CDSD implies larger differences in the terminal velocity of droplets. This is beneficial for collision coalescence and might cause the fast-rain process in the atmosphere (e.g., Göke et al., 2007). In addition, a broader CDSD increases the relative dispersion, which is the ratio of standard deviation to the mean CDSD. Previous studies show that an increase in relative dispersion is relevant to the albedo effect and can either increase or decrease albedo susceptibility depending on the broadening mechanism (Feingold et al., 1997; Liu and Daum, 2002; Feingold and Siebert, 2009). An interesting question is why the CDSD is wider than predicted; in particular, why large droplet sizes are frequently observed in the clouds (e.g., Siebert and Shaw, 2017). Several mechanisms have been proposed that can be divided into two categories: turbulence-induced spectra broadening and aerosol-induced spectra broadening. A brief review is given next for each category.

Turbulence is ubiquitous in the clouds and can cause CDSD broadening in both condensation and collision processes (e.g., Shaw, 2003; Devenish et al., 2012). Turbulence induces vertical oscillations of air parcels and causes fluctuations in temperature, water vapor concentration, and supersaturation (e.g., Ditas et al., 2012; Hammer et al., 2015). The effects of supersaturation fluctuations on droplet condensational growth in turbulent environments have been studied for several decades (e.g., Cooper, 1989; Khvorostyanov and Curry, 1999). A qualitative description of this mechanism is that some "lucky" cloud droplets experience relatively larger supersaturation or stay a relatively longer time in the cloud compared with the other cloud droplets; therefore they can grow larger in size and broaden the CDSD. Recent theoretical and experimental studies support this mechanism and provide ways to quantify the resulting width of the droplet size distribution (e.g., McGraw and Liu, 2006; Sardina et al., 2015; Chandrakar et al., 2016; Grabowski and Abade, 2017; Siewert et al., 2017). Turbulence can also modulate the condensational growth of cloud droplets through mixing and entrainment (e.g., Lasher-Trapp et al., 2005; Cooper et al., 2013; Korolev et al., 2013; Yang et al., 2016). In addition, turbulence can enhance the collision efficiency between droplets and produce "lucky" cloud droplets through stochastic collisions, which has been confirmed by direct numerical simulations and Lagrangian drop models (e.g., Paluch, 1970; Kostinski and Shaw, 2005; Falkovich and Pumir, 2007; Grabowski and Wang,

2013; Naumann and Seifert, 2015; de Lozar and Muessle, 2016).

Aerosols, which serve as condensation nuclei of cloud droplets, can also cause CDSD broadening in turbulent environments through several mechanisms. First, turbulence-induced mixing and entrainment can trigger in-cloud activation of haze parti-
cles, which can broaden the left branch of size distribution (e.g., Khain et al., 2000; Devenish et al., 2012; Yang et al., 2016; Grabowski et al., 2018). Secondly, giant cloud condensational nuclei (GCCN, usually defined as aerosols with dry diameter larger than a few $\mu m$) provides an embryo for large droplets, which can broaden the right branch of size distribution and can be important for warm rain initiation (e.g., Johnson, 1982; Feingold et al., 1999; Yin et al., 2000; Jensen and Lee, 2008; Cheng et al., 2009). Recently, Jensen and Nugent (2017) investigated the effect of GCCN on droplet growth and rain formation using a
cloud parcel model. They found that GCCN provides an embryo for big droplets at the activation stage and, more importantly, GCCN enhances droplet growth after activation due to the solute effect. For example, droplets formed on GCCN can still grow through the condensation of water vapor in the downdraft region even though the environment is subsaturated with respect to pure water (Jensen and Nugent, 2017). This, in fact, is an extreme case of Ostwald ripening.

Ostwald ripening for cloud droplets is the phenomenon when larger droplets grow and smaller droplets shrink due curvature and/or solute effects and, thus, it can broaden the CDSD at both small and large sides of the distribution. Srivastava (1991) investigated the growth of cloud droplets in a rising air parcel. Results show that the variance of squared radius of the CDSD was constant during the condensational growth process if both curvature and solute effects were ignored, but it was increased if those effects were considered. This "condensational broadening" is more pronounced in clouds with high cloud droplet
number concentration and low vertical velocity. In turbulent clouds, droplets will experience supersaturated/subsaturated conditions in updraft/downdraft regions. Korolev (1995) studied the evolution of the CDSD driven by supersaturation fluctuations in a vertically oscillating air parcel. Supersaturation fluctuations in his study mean that air is supersaturated in the updraft and subsaturated in the downdraft; however no spatial inhomogeneity of supersaturation is considered in the parcel. Results show that the growth and evaporation cycles during the CDSD evolution are irreversible if the solute and curvature effects are considered. This "CDSD irreversibility" (terminology used in his paper) will promote the growth of large cloud droplets, lead to evaporation or even deactivation of small cloud droplets, and thus broaden the CDSD. Korolev (1995) argued that stronger turbulent fluctuations of supersaturation would result in a broader CDSD. This is contrary to Çelik and Marwitz (1999), who found that supersaturation fluctuations are not responsible for CDSD broadening and the formation of large droplets. The curvature and solute effects on Ostwald ripening, activation and deactivation have been the topics of study in recent years (e.g.,
Wood et al., 2002; Arabas and Shima, 2017; Chen et al., 2018; Sardina et al., 2018) but, to our knowledge, the relative roles of the curvature effect and solute effect on CDSD broadening have not been investigated.

Here we consider an adiabatic cloud parcel that experiences vertical oscillations, with cloud droplets that are formed on polydisperse, sub-micrometer aerosols. Results confirm that the CDSD is broadened during diffusional growth due to Ostwald
ripening and associated droplet deactivation and reactivation, which is consistent with previous studies (e.g., Korolev, 1995;

Çelik and Marwitz, 1999). In this study, we investigate (1) what are the relative roles of the solute and curvature effects on CDSD broadening, and (2) what other factors can affect this broadening? This paper is organized as follows. Section 2 introduces the basic setup for cloud parcel model, which is similar to Jensen and Nugent (2017) except that there are no GCCN. Results related to CDSD broadening and the associated sensitivity studies are detailed in Section 3. Conclusions are summarized in Section 4, including a discussion of implications in cloud observations and modeling.

## 2 Methods

Historically there are two types of bin microphysics: fixed-bin scheme and moving-size-grid scheme (see section 4.2.1 in Khain et al. (2015) and references therein). The advantage of the moving-size-grid method is that it can avoid artificial CDSD broadening. In this study, we use a cloud parcel model with a moving-size-grid microphysics scheme, where discrete particle sizes on a 1-D grid (initially the radii of dry aerosols [e.g., Table 1]) each grow/shrink according to the environmental conditions to modify the 'moving-size' of the grid element. The original version of the model was designed to study cirrus clouds by Heymsfield and Sabin (1989), and then the warm clouds (Feingold and Heymsfield, 1992; Feingold et al., 1998). In recent years, this model has been modified and applied to investigate various of microphysical problems (e.g., Feingold and Kreidenweis, 2000; Xue and Feingold, 2004; Ervens and Feingold, 2012; Yang et al., 2012; Li et al., 2013; Yang et al., 2016). In the current version of parcel model, air pressure ($p$), parcel height ($h$), air temperature ($T$), water vapor mixing ratio ($q_v$), and radii of haze and cloud droplets ($r_i$) are prognostic variables, which are calculated using the variable-coefficient ordinary differential equation solver (VODE) (Brown et al., 1989). Specifically, $p$ is calculated from hydrostatic equation and $h$ depends on the vertical velocity ($w$). Similar to Eq. 11 in Heymsfield and Sabin (1989), $T$ is calculated from,

$$\frac{dT}{dt} = -\frac{g}{c_{p,air}}w + \frac{l_v}{c_{p,air}}\frac{dq_w}{dt}, \tag{1}$$

where $g$ is the gravitational acceleration, $c_{p,air}$ is the heat capacity of air, $l_v$ is the latent heat of water vaporization, and $q_w$ is the liquid water mixing ratio. The first term in Eq. 1 is the cooling due to dry adiabatic ascent, and the second term is the microphysical contribution due to the release of latent heat of condensation. Because the total water mixing ratio is conserved in the parcel, a decrease in water vapor mixing ratio ($-dq_v$) equals an increase in liquid water mixing ratio ($dq_w$). Air supersaturation ($S_e$), which controls the growth of haze and cloud droplets, is calculated from $T$, $p$ and $q_v$. A brief introduction of the model setup and the main mathematical formulations used for cloud microphysical processes are described below.

In this study, the parcel starts rising at about $300\ m$ below cloud base and starts descending at about $300\ m$ above cloud base, which is similar to Jensen and Nugent (2017), except that our cloud parcel then experiences upward and downward oscillations between $50\ m$ above cloud base and $300\ m$ above cloud base (see Figure 1a). The ascending and descending velocities are set to be $0.5\ m\ s^{-1}$ and $-0.5\ m\ s^{-1}$ for the control case. At the parcel's initial altitude of 600 m, the initial air temperature is $284.3\ K$, pressure is $938.5\ hPa$, and saturation ratio is $0.856$, which are as same as Jensen and Nugent (2017).

The initial dry aerosols are ammonium sulfate with a log-normal size distribution range of $10\ nm$ to $500\ nm$ in radius. The sub-micrometer aerosols are parsed into 100 grids (discrete droplet size in each grid detailed in Table 1), where the median radius is $50\ nm$ and the geometric standard deviation is 1.4. The total number mixing ratio is $1000\ mg^{-1}$ for the control case, which is about $1000\ cm^{-3}$ (see Figure 1b). The model first calculates the equilibrium size of haze droplets for each grid at
85.6% relative humidity, as does Jensen and Nugent (2017). The equilibrium size of haze particles for the $i$th grid ($r_i$) at initial relative humidity is obtained by solving the equation $S_{sat}(r_i) = RH(t = 0)$ iteratively, where $S_{sat}$ is the saturation ratio for a solution droplet, calculated from Köhler equation (Pruppacher and Klett, 2010, p. 172),

$$S_{sat} \equiv \frac{e}{e_s(T)} = a_s(r_{d,i}, r_i) \exp\left(\frac{2\sigma_s}{\rho_w R_v T r_i}\right), \tag{2}$$

where $e$ is the water vapor pressure in air, $e_s$ is the saturated water vapor pressure over a solution droplet at $T$, $\rho_w$ is the
density of water, and $R_v$ is the gas constant for water vapor. $\sigma_s$ is the water activity of the haze droplets, which is a function of temperature and solute (Pruppacher and Klett, 2010, p. 133). $a_s$ is the water activity of haze droplets, which depends on the composition of aerosol, size of dry aerosol ($r_d$), and size of haze droplets ($r$). In this study, $a_s$ for cloud droplets is calculated from laboratory-based parameterizations (Eq. 2 in Tang and Munkelwitz (1994)).

Only diffusional growths of haze and cloud droplets are considered in our model. Collision coalescence, sedimentation, mixing, and entrainment are ignored. The growth of haze or cloud droplet for the $i$th grid is calculated from,

$$\frac{dr_i}{dt} = \frac{1}{r_i} \frac{S_e - S_{sat}}{G}, \tag{3}$$

where $G$ is the growth parameter given by,

$$G = \left[\frac{\rho_w R_v T}{D_v' e_s(T)} + \frac{\rho_w l_v}{k_T' T}\left(\frac{l_v}{R_v T} - 1\right)\right]. \tag{4}$$

$D_v'$ and $k_T'$ are, respectively, the modified diffusion coefficient and the modified thermal diffusion coefficient (Lamb and Verlinde, 2011, p. 337-338),

$$D_v' = \frac{D_v}{\frac{r_i}{r_i + \lambda} + \frac{4D_v}{\alpha_m \bar{c}_{air} r_i}}, \tag{5}$$

and

$$k_T' = \frac{k_T}{\frac{r_i}{r_i + \lambda} + \frac{4k_T}{\alpha_T \bar{c}_{air} n_{air} c_{p,air} r_i}}. \tag{6}$$

Here $D_v$ is the physical diffusion coefficient, $k_T$ is the thermal diffusion coefficient, $\lambda$ is the mean free path of air, $\bar{c}_{air}$ is the mean molecular speed of air, and $n_{air}$ is the number concentration of air. $\alpha_m$ is the mass accommodation coefficient and $\alpha_T$ is the thermal accommodation coefficient. In this study, we choose $\alpha_m = 1.0$ and $\alpha_T = 1.0$.

$S_{sat}$ in the growth equation (Eq. 3) is calculated from the Köhler equation (Eq. 2). Therefore, the curvature effect (expo-
nential part in Eq. 2) and the solute effect ($a_s$ in Eq. 2) are considered during the growth process for each grid. It should be

noted that there are several methods to calculate the solute effect with the relative deviations for activation ranging up to $20\%$, but the differences are small for droplet growth (Pöschl et al., 2009). In addition, different choices of parameters–such as $\sigma_s$, $\alpha_m$ and $\alpha_T$–can also cause differences in droplet growth (Kreidenweis et al., 2003). How the choices of different parameters would affect our results is worth studying in the future. The total simulation time is 3 hours, and variables are recorded every 1 $s$ that include temperature, pressure, height, water vapor mixing ratio, as well as droplet size and number concentration for each grid.

## 3 Results and discussions

### 3.1 Cloud droplet size distribution broadening

For the control case, the liquid water mixing ratio increases linearly with height in the ascending branches and decreases in the descending branches as shown in Figure 2a. Liquid water mixing ratio in the ascending branch is slightly smaller than that in the descending branch at the same height due to the kinetic effect (or hysteresis effect), which is consistent with Korolev et al. (2013). The saturation ratio has an increasing trend in the ascending branch after each cycle, but has a decreasing trend in the descending branch (indicated by red and blue arrows in Figure 2b). Droplet size for two moving-size grids is shown in Figure 2c. Droplet size in the grid monotonically increases with the dry aerosol mass associated with the grid. The solid line is for the cloud droplet that formed on a dry aerosol of 503 $nm$ and represents the largest droplet in our simulation. It grows in the ascending branch but it evaporates in the descending branch. Also, the droplet size for this grid increases after each cycle. The dashed line in Figure 2c is for the cloud droplet that formed on a dry aerosol of 51 $nm$. For this cloud droplet, the changes in radius with height are similar for the initial few cycles, after which the cloud droplet deactivates and becomes a haze particle. Ultimately, the aerosol is reactivated again as a cloud droplet by the end of the simulation (green dashed line). Also notice that a second mode appears in the CDSD due to reactivation of aerosols after about 2 hours (see Figure 2d). It should be mentioned that the critical radius, where the Köhler curve peaks and a droplet is activated, is 3.6 $\mu m$ for a cloud droplet formed on a dry aerosol of 503 $nm$, and 0.44 $\mu m$ when formed on a dry aerosol of 51 $nm$. Figure 2d shows that all droplet radii are larger than 4 $\mu m$ at the end of updraft cycle, indicating that all cloud droplets are activated at that point. Because GCCN do not exist in our simulation and the oscillation frequency is low, all cloud droplets have enough time to grow to be activated in the updraft region. In this study, we focus on the CDSD at the end of the updraft cycle so the growth and evaporation of unactivated cloud droplets (e.g., McFiggans et al., 2006) will not affect the final CDSD. The CDSD broadens after each cycle as the larger droplets become larger and the smaller droplets either remain similarly sized or become smaller. All these features are consistent with Korolev (1995) (see Fig. 5 in his paper).

Korolev (1995) analytically investigates the narrowing and broadening of cloud droplet size distribution during condensation when solute and curvature effects are considered. He considers a cloud parcel oscillating vertically in simple harmonic motion. Results show that the CDSD evolution is irreversible if solute and curvature effects are considered. Irreversibility of

the CDSD will not only promote the growth of large droplets, but it will also lead to the evaporation, or even deactivation of small cloud droplets, and thus broaden the CDSD. However, the relative roles of the solute effect, curvature effect, deactivation and reactivation on the broadening of droplet size distributions have not been investigated.

To explore the relative roles of different factors in this CDSD broadening mechanism, three more cases are tested here. For the first case, we turn off both the solute and curvature effects for all cloud droplets after $700\ s$; this is the time when the cloud parcel first reaches $50\ m$ above cloud base and is just below the oscillation layer. Specifically, we set $S_{sat} = 1$ for all droplets. The result is shown in Figure 3a. For this case, the CDSD repeats for each cycle, consistent with Korolev et al. (2013), and the total cloud droplet number concentration ($n$) is constant (red solid line in Figure 3d). For the second case, we only turn off the

curvature effect but retain the solute effect. Specifically, we ignore the exponential term in Eq. 2 such that $S_{sat} = a_s$. The result in Figure 3b shows that the largest droplet (with the most solute) can grow after each cycle while the smallest droplet size (with the least solute amount) associated with a moving-size grid does not change much after each cycle. However the largest droplet size that a grid can reach is much smaller than that in the control case. Because the saturated water vapor pressure over a droplet formed on larger aerosol is lower than that formed on smaller aerosol due to the solute effect, the larger droplet grows faster than the smaller droplet in the updraft region, and it evaporates slower in the downdraft region. For this case, the solute effect

alone cannot explain the larger cloud droplets in the control case. In addition, $n$ is also a constant and droplet deactivation does not occur (green dashed line in Figure 3d). In the third case, we consider both curvature and solute effects, but we do not allow droplet reactivation. This means that once the droplet deactivates it cannot be activated again. The result in Figure 3c shows that the growth of the largest cloud droplet is similar to the control case, but the size of the smallest cloud droplet associated with a grid also increases after each cycle. The reason for this CDSD broadening is the Ostwald ripening effect, where large

droplets grow at the expense of small ones. Past studies have concluded that the ripening effect is typically slow and inefficient for droplet growth (Wood et al., 2002). But the vertical oscillations near cloud base that are considered here allow for droplet deactivation and result the decrease of $n$ with time (see Figure 3d), as in the control case. Thus, the typically inefficient Ostwald ripening is amplified through the resulting deactivation of the smallest droplets. An early suggestion of this behavior is shown

in Fig. 8 of Hagen (1979). The only difference between the control and this simulation is that $n$ for the control case increases near the end of the simulation because of droplet reactivation (see Figure 3d). It should be mentioned that the step changes in $n$ in Figure 3d are a result of using a discretized grid method to represent the continuous spectrum. A downward step in $n$ means droplet deactivation, and an upwards step in $n$ means droplet reactivation. Deactivation and reactivation can also be seen from the CDSD qualitatively: droplet deactivation occurs when the peak value of CDSD decreases (from red to blue as shown in

Figure 2d), while droplet reactivation occurs when a subset of smaller cloud droplets appears.

From Figures 3 a and b, we can see that the solute effect contributes part of the CDSD broadening compared with the control case. But the solute effect alone is not enough to explain the growth of the largest cloud droplet. Droplet deactivation, which is related to the curvature effect, plays a crucial role here (see Figure 3c). Because the oscillations occur within the cloud re-

gion, $50\ m$ above cloud base, droplet deactivation is surprising to us. There are two related questions: (1) Why do some cloud

droplets deactivate in the cloud region while others do not? (2) Why is droplet deactivation related to the CDSD broadening?

The reason for the droplet deactivation is mainly because the cloud parcel experiences upwards and downwards oscillations. In the downdraft region, the air is subsaturated, which supports droplet evaporation. In addition, the saturated water vapor pressures over polydisperse droplets are different via both the solute and curvature effects. Smaller droplets with less solute and larger radii of curvature have higher saturated water vapor pressures, and thus evaporate faster than larger droplets in the downdraft region. Therefore, smaller droplets will evaporate first in the downdraft region.

The reason why droplet deactivation is related to the CDSD broadening can be explained in two ways. From the thermodynamic point of view, the liquid water mixing ratio is roughly a constant at a given height for each cycle (see Figure 2a). As the $n$ decreases due to the droplet deactivation, we can expect that on average droplet size will be larger because the same amount of water will be redistributed on fewer cloud droplets. From the kinetic point of view, quasi-steady state supersaturation ($s_{qs}$) will become larger after each cycle due to droplet deactivation, as shown in Figure 2b. $s_{qs}$, the environmental supersaturation in quasi-steady state, is inversely proportional to the integral of mean droplet size $\overline{r}$ and droplet number concentration ($n$), $s_{qs} \propto (\overline{r}n)^{-1}$ (e.g., Squires, 1952; Politovich and Cooper, 1988; Korolev and Mazin, 2003; Lamb and Verlinde, 2011). Here the decrease in $n$ due to droplet deactivation is much greater than the change of $\overline{r}$; therefore, $s_{qs}$ will increase with decreasing $n$. This means that larger droplets grow even faster in the updraft region, and smaller droplets evaporate even faster in the downdraft region – beyond the solute effect alone. Conversely, an increase in $s_{qs}$ will enhance droplet deactivation for smaller droplets, and it will also reinforce the growth of larger droplets in a positive feedback.

One question relevant to precipitation initiation is how fast can the largest cloud droplet grow in an oscillating parcel compared with droplets in an ascending-only parcel? For the latter case, the cloud parcel ascends at a vertical velocity of $0.5\ m\ s^{-1}$ for three hours with the same initial condition as the control case. At the end of the simulation, the cloud parcel reaches about $6000\ m$ and cloud droplets are supercooled (around 248 K), but we ignore ice nucleation in this study. The mean (yellow dashed line) and largest/smallest (upper/lower gray dashed lines) cloud droplets in an ascending-only cloud parcel are also shown in Figure 2d. It can be seen that the size of the largest cloud droplet in a moving-size grid at cloud top in each cycle of the oscillating parcel (blue color bar) is similar to that in the ascending-only parcel (upper gray line). This is quite surprising because when the parcel reaches $1200\ m$ for the first time (i.e., the top of the oscillation cycle), the largest cloud droplet radius is $9.07\ \mu m$ (see Table 2 and Figure 2c); however after several cycles, the largest cloud droplet radius is $17.3\ \mu m$, still at 1200 $m$. The size is similar to the largest droplet size associated with a moving-size grid in an ascending-only parcel at a height of about $6000\ m$. This means that the largest cloud droplet size for a grid in an oscillating parcel at $1200\ m$ is much larger than calculated from a traditional cloud parcel model (ascent only), and hence shows "superadiabatic" growth. In addition, the size of the smallest cloud droplet for a grid and the mean droplet size are larger in an ascending-only parcel. Differences between the mean droplet sizes increases after each cycle, especially at the end of the simulation due to the reactivation of numerous small droplets. Therefore, the relative dispersion, which is the ratio of the standard deviation to the mean of a droplet size

distribution, also increases after each cycle, and is much larger than in an ascending-only cloud parcel.

## 3.2 Sensitivity studies

In this subsection, we investigate effects of several factors on the CDSD in the adiabatic parcel model with vertical oscillations.
Previous studies show that aerosol number concentration and vertical velocity are the two most important factors controlling cloud properties in an adiabatic cloud parcel model (e.g., McFiggans et al., 2006; Reutter et al., 2009; Chen et al., 2018). Two regimes are frequently considered: an aerosol-limited regime exists when there is ample supply of water, and the cloud droplet number concentration is limited by the aerosol number concentration; and an updraft-limited regime exists when supersaturation is starved, and the cloud droplet number concentration is limited by the updraft velocity. In the updraft-limited region, cloud droplets will compete with each other for the limited available water, and the larger aerosols will suppress the activation of smaller aerosols (Ghan et al., 1998; Feingold and Kreidenweis, 2000; Feingold et al., 2001). Based on Reutter et al. (2009), the aerosol-limited regime exists when the ratio of the vertical velocity to droplet number concentration, $w/n$, is larger than $10^{-3} \ ms^{-1}cm^3$ and the updraft-limited region occurs when the $w/n$ ratio is smaller than $10^{-4} \ ms^{-1}cm^3$. For the control case, the $w/n$ ratio is $7 \times 10^{-4} \ ms^{-1}cm^3$, which is in the transitional regime. In this subsection, we choose several values of aerosol number concentration and vertical velocity to investigate the CDSD in the aerosol-limited and updraft-limited regimes. In this subsection, we choose several values of aerosol number concentration and vertical velocity to investigate the CDSD in the aerosol-limited and updraft-limited regimes. In addition, we also test the effect of the recirculation layer thickness on the CDSD broadening.

### 3.2.1 Effect of total aerosol number concentration

We test two other aerosol number concentrations, $10^2 \ cm^{-3}$ and $10^4 \ cm^{-3}$, and keep the median radius and geometric standard deviation the same as the control case (see Figures 4 a and c ). These values are chosen to represent the conditions for clean clouds ($10^2 \ cm^{-3}$) and polluted clouds ($10^4 \ cm^{-3}$), which are consistent with previous studies (e.g., Xue and Feingold, 2004; Chen et al., 2018). Considering a vertical velocity of $0.5 \ ms^{-1}$, they also represent the aerosol-limited regime (the $10^2 \ cm^{-3}$ case leads to a $w/n$ ratio of $5 \times 10^{-3} \ ms^{-1}cm^3$) and the transition regime (the $10^4 \ cm^{-3}$ case leads to a $w/n$ ratio of $4 \times 10^{-4} \ ms^{-1}cm^3$). The results show that the CDSD for the relatively clean case ($10^2 \ cm^{-3}$) behaves similarly to the solute effect alone (compare Figures 3b and 4b) – there is neither droplet deactivation nor reactivation. The CDSD broadening is due to the ripening effect alone, which is not as efficient as when it is accompanied by deactivation as in the control case. For the relatively polluted case ($10^4 \ cm^{-3}$), both droplet deactivation and reactivation occur (see Figure 4d). The largest cloud droplet acts similarly as that in the control case, while the smallest cloud droplet is larger 1.5 h into the simulation but then begins to become smaller compared with the control case. We interpret these observations as follows. For the clean case, all aerosols are activated, and all droplets are able to grow to a relatively large size, making them unlikely to deactivate. However for polluted case, not all CCN are activated, there are therefore some smaller droplets that cannot grow very large and they will evaporate

first in the downdraft region. Another explanation from Korolev (1995) is that the CDSD broadening occurs when air super-saturation ($S_e$) is smaller than the critical supersaturation for the smallest cloud droplets ($S_{sat}(r_{small})$). For this condition, the smallest cloud droplets evaporate and the largest cloud droplets might grow slightly if $S_e > S_{sat}(r_{large})$ or evaporate slightly if $S_e < S_{sat}(r_{large})$, thus leading to broadening. If the water vapor mixing ratio in air on average is much larger than the satu-

rated water vapor mixing ratio over droplet, only narrowing of the CDSD occurs. Because in-cloud supersaturation decreases with increased aerosol concentration, it is expected that the Ostwald ripening is more efficient in polluted cloud, which is also consistent with (Srivastava, 1991).

### 3.2.2    Effect of vertical velocity

Two vertical velocities ($0.1\ m\ s^{-1}$ and $1.0\ m\ s^{-1}$) are used to test their influence on CDSD broadening. These values are chosen based on observations that updrafts in stratocumulus clouds are on the order of $0.1\ m\ s^{-1}$ and in cumulus clouds are on the order of $1.0\ m\ s^{-1}$ (Ditas et al., 2012; Katzwinkel et al., 2014). Results also show that they correspond to the aerosol-limited regime (the $1.0\ m\ s^{-1}$ case leads to a $w/n$ ratio of $10^{-3}\ ms^{-1}cm^3$) and the transitional regime (the $0.1\ ms^{-1}$ case leads to a $w/n$ ratio of $5 \times 10^{-4}\ ms^{-1}cm^3$). For a relative low velocity of $\pm 0.1\ m\ s^{-1}$, the cloud parcel only experiences one

and a half cycles within three hours (see Figure 5a). The parcel reaches cloud base around 1 hour, significantly later than the control case due to the small velocity (see Figure 5a). However, the largest cloud droplet size ultimately becomes similar to that in the control case, and we also see the cloud droplet number concentration decrease due to droplet deactivation. No droplet reactivation occurs because the small velocity generates a low supersaturation in the updraft region, which is unfavorable for droplet reactivation. For a relative high velocity of $\pm 1.0\ m\ s^{-1}$, the cloud parcel can cycle more times within three hours (see

Figure 5c). The parcel reaches cloud base faster than the control case (see Figure 5c). Here we keep the thickness of the recirculation layer constant. Therefore, larger vertical velocity results in a higher oscillation frequency. Both droplet deactivation and reactivation occur in this case, and the largest and smallest cloud droplets behave similarly to the control case.

### 3.2.3    Effect of the thickness of recirculation layer

Turbulence driven by cloud-top radiative cooling can result in various eddy sizes in the stratocumulus-topped boundary layer (Wood, 2012). Two different recirculation layer depths are tested, $150\ m$ and $350\ m$, to investigate the effect of eddy size on CDSD broadening. For a recirculation layer of $150\ m$, which is $100\ m$ thinner than the control case, the parcel experiences more cycles within three hours (see Figure 6a). The total cloud droplet number concentration decreases with time due to droplet deactivation, but no droplet reactivation occurs (see Figure 6b). Therefore the largest cloud droplet is similar to the control case,

but the smaller cloud droplet is larger than in the control case. For a recirculation layer of $350\ m$, the parcel can penetrate the cloud base each cycle (see Figure 6c). In this case, all cloud droplets are deactivated below cloud base and reactivated again when the cloud parcel is supersaturated in the next ascending branch. Therefore the CDSD is repeated and no broadening

occurs.

## 3.3 Discussion

We have studied the effects of total aerosol number concentration, updraft velocity, and thickness of the recirculation layer on
CDSD broadening. However we note that there are other parameters used in this study that can lead to the uncertainties in the
results. For example, Takeda and Kuba (1982) found that using an insufficient number of model grids will lead to the narrow
CDSD reported by Mordy (1959). Kreidenweis et al. (2003) found that both the spectral discretisation and the uncertainty in
the value of mass accommodation coefficient can lead to uncertainty in the results. To test the effects of mass accommoda-
tion coefficient and spectrum discretization on the CDSD, two more sensitivity studies are conducted. One case is to set mass
accommodation coefficient ($\alpha_m$) to 0.06 based on Shaw and Lamb (1999). It is expected that a smaller value of $\alpha_m$ might
suppress the growth of cloud droplets. The other case is to change the number of grids from 100 to 200, while keeping other
parameters the same as in the control case.

Table 2 summarizes the microphysical properties at cloud top for different cases. When the cloud parcel first reaches about
1200 $m$, the largest cloud droplet radius associated with a moving-size grid ($r_{max}$) is 9.1 $\mu m$ (case 0). If the cloud parcel
continues rising for three hours as for the ascending-only case, $r_{max} = 17$ $\mu m$ at 6000 $m$. However if the parcel experiences
recirculation within cloud region, $r_{max}$ can also be around 17 $\mu m$ as long as deactivation occurs, except for the low $N_a$ case
(see Table 2). If reactivation also occurs, the smallest cloud droplet radius associated with a moving-size grid $r_{min}$ is around 5
$\mu m$ and the relative dispersion is larger than 0.1. It is interesting to note that low mass accommodation has a negligible effect
on $r_{max}$, but it has a stronger impact on $r_{min}$. This will result in a broader CDSD compared with the control case. In addition,
a low mass accommodation coefficient inhibits the growth of cloud droplets and leads to more activated cloud droplets (Xue
and Feingold, 2004). Results for 200 grids are similar to that for the control case, which means that the 100 grids used in this
study are enough to limit the uncertainty due to spectrum discretization.

From the above, we see that droplet deactivation and droplet reactivation play crucially important roles in CDSD broadening
in this study. Deactivation of smaller droplets is important for the growth of larger cloud droplets (e.g., see Figures 2d, 3c, 4d,
5b,d and 6b). Droplet deactivation occurs in the descending branch for smaller droplets due to both the curvature and solute
effects (Ostwald ripening). The evaporation of smaller cloud droplets with less solute makes water vapor available for the
growth of other larger cloud droplets. On average, the largest cloud droplet size for a moving-size grid increases with time after
each cycle.

Results from the sensitivity studies show that the relative dispersion is larger than 1.5 for relatively polluted conditions when
both deactivation and reactivation occur (see Table 2), which is consistent with the values from observations and simulations
(e.g., Miles et al., 2000; Liu and Daum, 2002; Chandrakar et al., 2016). However the relative dispersion has also been found to

be larger than 1.5 for relatively clean conditions (e.g., Miles et al., 2000; Lu and Seinfeld, 2006; Chandrakar et al., 2016). This might be due to other mechanisms, such as supersaturation fluctuations in a turbulent environment or the collision coalescence process. It should be mentioned that the CDSD observed in previous studies might have the problem of instrumental broadening due to low instrument resolution or long-distance averaging of the sampling volume (Brenguier et al., 2011; Devenish et al., 2012). A broad CDSD is also observed by recent holographic measurements, which limit the effect of instrument broadening and have much higher temporal and spatial resolution than other instruments, such as particle-counting probes (Beals et al., 2015; Glienke et al., 2017; Desai et al., 2018).

We note that deactivation is suppressed for a thin recirculation layer $\Delta H = 150\ m$ as shown in Figure 6b, and therefore the CDSD broadening is not as efficient as the control case. However, the vertical oscillations of an air parcel due to turbulence might be much smaller than $150\ m$. Wood et al. (2002) did not observe the enhanced CDSD broadening by deactivation and reactivation with a shallower recirculation layer. One interesting question is whether deactivation or reactivation be inhibited for a very thin recirculation layer. To answer this question, three more cases are carried out with recirculation layers of $50\ m$, $5$ $m$ and $1\ m$. All these cases have the same setup as the control case except for the thickness of recirculation layer. The CDSD and total cloud droplet number concentration for each case are shown in Figure 7. It can be seen that reactivation is inhibited for all cases, but deactivation always occurs. More interestingly, the CDSD for all these three cases are similar, and the decrease of total cloud droplet number concentration due to deactivation is also similar. The evolution of the CDSD for a thin recirculation layer is independent of air motion and degrades to a steady state where the CDSD broadening is due to Ostward ripening in a still environment.

One interesting result is that the size of the largest cloud droplet associated with a moving-size grid within each cycle is similar to that in the ascending-only parcel (i.e., approximately within one micrometer), as shown in Figure 8. The general trends approximately follow the growth rate that is independent of aerosol number concentration, vertical velocity and the thickness of the oscillation layer, as long as deactivation occurs. This suggests that the growth of the largest cloud droplets strongly depends on the amount of time such droplets remain in the cloud (residence time of cloud droplets), rather than the temporal variability of supersaturation in updrafts and downdrafts. The reason is that the environmental (i.e., the in-cloud) saturation ratio ($S_e$) is buffered by the equilibrium saturation ratio ($S_{sat}$) over smaller droplets. Figure 9 shows the changes of $S_e$ and $S_{sat}$ over two droplets (same used as in Figure 2c) in the control case. Instead of being symmetric around $1$ for the pure water case (ignoring solute and curvature effects), $S_e$ in the oscillating parcel is symmetric around $S_{sat}$ over the small cloud droplets. For example before 1.5 hours, droplets formed on $r_a = 51nm$ are the smallest cloud droplets in the population, and the average $S_e$ (gray line) during one oscillation is roughly symmetric around the blue line (Figure 9). The fact that $S_e$ is buffered by $S_{sat}$ over small cloud droplets is mainly because the number concentration of the smallest cloud droplet ($36\ cm^{-3}$ in the control case) is much larger than that of large cloud droplet ($1.8 \times 10^{-9}\ cm^{-3}$). When those small droplets deactivate (between 1.5 to 2.5 hours), $S_{sat}$ (blue line) for those deactivated droplets is the same as $S_e$ (gray line). During this period, $S_e$ is symmetric around $S_{sat}$ over the remaining small droplets (larger than the droplets formed on $r_a = 51\ nm$ but smaller than for $r_a = 503\ nm$).

When the droplets formed on $r_a = 51\ nm$ are reactivated (after 2.5 hours), $S_e$ is symmetric around $S_{sat}(r_a = 51\ nm)$ again until they are deactivated. It should be mentioned that number concentration of those reactivated droplets increases steady after each cycle after 2.0 hours (See Figure 3d). By the end of the simulation, the number concentration of the reactivated droplets is similar to that of the remaining large droplets (about $150\ cm^{-3}$). Therefore, the effect of those reactivated droplets on the environmental saturation ratio becomes stronger after 2.0 hours (see Figure 9).

This symmetric property of $S_e$ can be also explained using the quasi-steady supersaturation $s_{qs}$. For pure water droplets, $s_{qs} \sim \frac{w}{nr}$ (Lamb and Verlinde, 2011). This can be obtained from the analytical expression of supersaturation in an adiabatic cloud parcel: $\frac{dS_e}{dt} = Aw - Bnr(S_e - 1)$, where A and B are parameters depending on thermodynamic properties (Korolev and Mazin, 2003). A symmetric distribution of $w$ around 0 will generate a symmetric distribution of $s_{qs}$ around 0 (i.e., $S_e$ around 1). If curvature and solute effects are considered, $s_{qs}$ will be symmetric around $s_k$ given the same condition of $w$, because $\frac{dS_e}{dt} = Aw - Bnr(S_e - S_{sat})$ and thus $s_{qs} \sim \frac{w}{nr} + s_k$, where $s_k = S_{sat} - 1$ is the equilibrium supersaturation ratio over a mono-disperse droplet. In the updraft region, all droplets grow and the effect of $s_k$ is negligible. In the downdraft region and for polydisperse cloud droplets, the large number of small cloud droplets buffers the environmental conditions. Therefore $S_e$ is symmetric around $S_{sat}$ over smaller droplets before they deactivate in the oscillating parcel. $\overline{S_e} - S_{sat}$ controls the growth of a large droplet and it is positive on average. That is why the large droplets can grow after each cycle. In addition, the influence of $S_e$ fluctuations on droplet growth is small if $S_{sat}$ over a large droplet is much lower than $S_e$ and its fluctuations. The extreme examples of this phenomenon are when droplets form on GCCN in warm clouds (Jensen and Nugent, 2017) or ice particles form in mixed phase clouds. Therefore, the growth of the large droplet here is dominated by its in-cloud lifetime. Previous studies show that although the mean lifetime of cloud droplets is usually less than half an hour, the residence time for some lucky cloud droplets can be longer than one hour (e.g., Feingold et al., 1996; Kogan, 2006; Andrejczuk et al., 2008). Those long-lifetime cloud droplets might contribute to large droplets in the cloud, similar to long-lifetime ice particles in mixed-phase clouds (Yang et al., 2015).

However if all cloud droplets are deactivated, CDSD broadening does not occur (see Figure 6d). Without droplet deactivation, the CDSD can also broaden due just to the solute effect, as is the case when the curvature effect is ignored (Figure 3b) or when the total aerosol number concentration is low (Figure 4b). CDSD broadening due to the ripening effect without droplet deactivation is not as significant as it is with droplet deactivation, but it also might be important after several hours as suggested by Wood et al. (2002).

Droplet reactivation usually occurs in the updraft region after several cycles, and those reactivated droplets will be deactivated again in the downdraft region. Formation of smaller cloud droplets can broaden the CDSD at smaller sizes, decrease the mean cloud droplet size, and increase the relative dispersion. Meanwhile, the generation of new cloud droplets also suppresses the growth of larger cloud droplets (see Figure 2d).

In summary, the results of this study show that the CDSD can be broadened in a vertically oscillating cloud parcel if both solute and curvature effects are considered, consistent with the findings of previous studies (e.g., Korolev, 1995). Although our model uses an idealized setup, the sensitivity studies help explore the conditions under which this mechanism may be important in the real clouds. The results show that CDSD broadening due to Ostwald ripening can be enhanced in relatively polluted conditions when deactivation and reactivation occur, such as typically exists for continental clouds. For relatively clean conditions like marine clouds, other CDSD broadening mechanisms might be more relevant, such as the collision coalescence process or supersaturation fluctuations due to turbulence. When deactivation and reactivation occur, the simulation results show that the smallest cloud droplets do not change significantly after each oscillation cycle, while the largest cloud droplets grow on average after each cycle. The growth of the largest cloud droplet depends on its in-cloud lifetime. This is because, due to the solute effect, the saturation water vapor pressure over larger cloud droplets is smaller than the environmental water vapor pressure that is buffered by numerous smaller cloud droplets with smaller amounts of solute. It should be mentioned that the system is buffered by smaller cloud droplets formed on smaller CCN when the number concentration of those droplets is much more than that for the largest cloud droplets formed on the largest CCN. This may not be true under relatively clean conditions, where the environmental supersaturation can be affected by droplets formed on the largest CCN.

## 4 Conclusions and atmospheric implications

In this study, we investigate the condensation growth of cloud droplets in an adiabatic parcel with vertical oscillations based on a moving-size-grid cloud parcel model where cloud droplets are formed on polydisperse, sub-micrometer aerosol particles. Both the solute and curvature effects are considered for all cloud droplets before and after activation during the whole simulation. The CDSD can also broaden by condensation growth due to Ostwald ripening together with droplet deactivation and reactivation, which is consistent with the results of Korolev (1995). Droplet deactivation occurs in the descending branch due to the combination of the solute and curvature effects. Deactivation of smaller droplets makes water vapor available for other larger droplets, and thus broadens the CDSD at larger sizes. The growth of the largest cloud droplet in a vertically oscillating cloud parcel approximately follows the growth rate in an ascending-only cloud parcel after each cycle, and it is independent of aerosol number concentration, vertical velocity, and the thickness of the oscillation layer, as long as deactivation occurs. The size of the largest cloud droplet strongly depends on the time that droplet remains in the cloud rather than on the variability of the in-cloud supersaturation. This is because the large number of smaller cloud droplets buffers the environmental air: the environmental saturation ratio in an oscillating parcel is symmetric around the equilibrium saturation ratio over smaller cloud droplets. The growth rate for the largest cloud droplets can be used to roughly estimate the large-size upper boundary of the CDSD, at least in this study. Droplet reactivation usually occurs after a few cycles. These cloud droplets are activated in the ascending branch, and deactivated in the descending branch. They are usually very small (less than $5\ \mu m$) and thus broaden the CDSD at smaller sizes. The mean cloud droplet size significantly decreases when reactivation occurs, which leads to an increase in relative dispersion. On the other hand, those newly formed cloud droplets compete against other cloud droplets for

water vapor, thus suppressing the growth of larger cloud droplets.

We note that there are additional factors that might affect droplet growth that are not treated in this study. For example, we do not consider the sedimentation of cloud droplets in this study, similar to Korolev et al. (2013) and Jensen and Nugent (2017). This is a reasonable assumption for an updraft velocity of $0.5 \ m \ s^{-1}$ or above, but ignoring sedimentation in the low velocity case ($0.1 \ m \ s^{-1}$) will limit the accuracy of our results. In addition, we do not consider the collision coalescence between droplets. Although CDSD broadening is favorable for collision processes, it might be interesting to determine how this broadening will accelerate rain formation.

We have used idealized simulations to analyze the CDSD broadening in a vertically oscillating cloud parcel due to Ostwald ripening. There are three necessary conditions for this CDSD broadening mechanism. The first condition is that droplets form on polydisperse aerosol particles where larger cloud droplets contain more solute. This is a very general occurrence in the atmosphere due to the complexity of aerosol size and composition (Murphy et al., 1998; Khain et al., 2000). The second condition is that a cloud experiences upward and downward oscillations. This is also a general occurrence in natural clouds due to turbulence and circulations that can become established within a cloud layer (Wood, 2012). The third condition is that cloud droplets have a long in-cloud residence time, e.g., longer than 1 hour. This is consistent with previous studies that cloud droplet residence time plays an important role in CDSD broadening due to the Ostwald ripening effect (Wood et al., 2002; Romakkaniemi et al., 2009). We expect that this mechanism of CDSD broadening is possible in the real clouds under those specific conditions.

It should be mentioned that one limitation of this study arises from the use of the adiabatic assumption for three-hour simulations. Turbulence can result in not only upward and downward oscillations but also in entrainment and mixing (Shaw, 2003; Devenish et al., 2012). The latter can cause cloud droplet evaporation, deactivation and reactivation (Korolev et al., 2013; Yang et al., 2016). In addition, the lifetime of the cloud parcel is usually less than one hour (Andrejczuk et al., 2008). Therefore, one should be aware that results in this study are based on a very idealized state. More realistic studies should consider mixing processes where for example a trajectory ensemble model would be a suitable tool (Ovchinnikov and Easter, 2010; Feingold et al., 1998). How important this mechanism is to CDSD broadening in real clouds compared with other mechanisms is worth future investigation, but is beyond the scope of this study.

There is an implication of this mechanism for the cloud modeling community. Most of the bulk and bin microphysical schemes only consider the curvature and solute effects during the activation process based on Köhler theory. Cloud droplets are assumed to be pure water after they are activated. Tracking the solute distribution for each bin of cloud droplet is possible using a joint 2-D bin aerosol-cloud microphysical scheme, but it is very computationally expensive (e.g., Andrejczuk et al., 2010; Ovchinnikov and Easter, 2010; Lebo and Seinfeld, 2011). The mechanism of CDSD broadening in this study requires the model to consider both solute and curvature effects all the time (i.e., before and after activation, deactivation and reactivation).

Our results suggest the importance of solute and curvature effects to the deactivation and reactivation processes, which are consistent with previous studies (e.g., Andrejczuk et al., 2008; Hoffmann et al., 2015; Hoffmann, 2017; Chen et al., 2018). However the results are counter to some other studies where details of activation and deactivation are argued to be unimportant in the cloud simulation (e.g., Srivastava, 1991; Chuang et al., 1997; Grabowski et al., 2018). Large eddy simulations with a

5 similar microphysical treatment would be useful to investigate how important this mechanism is to CDSD broadening in more realistic clouds.

*Acknowledgements.* F.Y., P.K. and A.M.V. were supported by the U.S. Department of Energy (DOE) under Contract DE-SC0012704. R.A.S. was supported by the DOE Office of Science as part of the Atmospheric System Research program through grant no. DE-SC0011690.

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

**Table 1.** Initial dry aerosol radii for different grids.

| grid number | $r_{dry}$ (nm) | grid number | $r_{dry}$ (nm) | grid number | $r_{dry}$ (nm) | grid number | $r_{dry}$ (nm) |
|---|---|---|---|---|---|---|---|
| 1 | 503 | 26 | 191 | 51 | 72.4 | 76 | 27.5 |
| 2 | 484 | 27 | 184 | 52 | 69.7 | 77 | 26.4 |
| 3 | 466 | 28 | 177 | 53 | 67.0 | 78 | 25.4 |
| 4 | 448 | 29 | 170 | 54 | 64.5 | 79 | 24.5 |
| 5 | 431 | 30 | 163 | 55 | 62.0 | 80 | 23.5 |
| 6 | 414 | 31 | 157 | 56 | 59.7 | 81 | 22.6 |
| 7 | 399 | 32 | 151 | 57 | 57.4 | 82 | 21.8 |
| 8 | 384 | 33 | 146 | 58 | 55.2 | 83 | 20.9 |
| 9 | 369 | 34 | 140 | 59 | 53.1 | 84 | 20.2 |
| 10 | 355 | 35 | 135 | 60 | 51.1 | 85 | 19.4 |
| 11 | 341 | 36 | 130 | 61 | 49.2 | 86 | 18.6 |
| 12 | 328 | 37 | 125 | 62 | 47.3 | 87 | 17.9 |
| 13 | 316 | 38 | 120 | 63 | 45.5 | 88 | 17.3 |
| 14 | 304 | 39 | 115 | 64 | 43.8 | 89 | 16.6 |
| 15 | 292 | 40 | 111 | 65 | 42.1 | 90 | 16.0 |
| 16 | 281 | 41 | 107 | 66 | 40.5 | 91 | 15.4 |
| 17 | 271 | 42 | 103 | 67 | 39.0 | 92 | 14.8 |
| 18 | 260 | 43 | 98.8 | 68 | 37.5 | 93 | 14.2 |
| 19 | 250 | 44 | 95.0 | 69 | 36.0 | 94 | 13.7 |
| 20 | 241 | 45 | 91.4 | 70 | 34.7 | 95 | 13.2 |
| 21 | 232 | 46 | 87.9 | 71 | 33.4 | 96 | 12.7 |
| 22 | 223 | 47 | 84.6 | 72 | 32.1 | 97 | 12.2 |
| 23 | 214 | 48 | 81.4 | 73 | 30.9 | 98 | 11.7 |
| 24 | 206 | 49 | 78.3 | 74 | 29.7 | 99 | 11.3 |
| 25 | 198 | 50 | 75.3 | 75 | 28.6 | 100 | 10.8 |

**Table 2.** Microphysical properties at cloud top for different cases: $r_{max}$ is the largest cloud droplet radius in a moving-size grid, $r_{min}$ is the smallest cloud droplet radius in a grid, $\bar{r}$ is the mean cloud droplet size, $\sigma$ is the standard deviation of droplet radius, $\sigma/\bar{r}$ is the relative dispersion and $n$ is the cloud droplet number concentration. Case 0 is when the cloud parcel reaches the cloud top for the first time with the same setup as the control case (shown as black circle in Figure 3). For other cases, results represent the parcel at cloud top for the last time after 3 hours simulation; the example of the control case is shown as the green circle in Figure 3.

| | $r_{max}$ $(\mu m)$ | $r_{min}$ $(\mu m)$ | $\bar{r}$ $(\mu m)$ | $\sigma$ $(\mu m)$ | $\frac{\sigma}{\bar{r}}$ | n $(cm^{-3})$ | deactivation | reactivation |
|---|---|---|---|---|---|---|---|---|
| case 0 | 9.1 | 4.2 | 5.8 | 0.5 | 0.088 | 654 | no | no |
| ascending only | 17 | 12 | 13 | 0.55 | 0.041 | 654 | no | no |
| control | 17 | 6.1 | 7.5 | 1.6 | **0.22** | 260 | **yes** | **yes** |
| $\alpha_m = 0.06$ | 17 | 5.1 | 7.0 | 1.9 | **0.27** | 299 | **yes** | **yes** |
| $N_{grid}$=200 | 17 | 5.9 | 7.5 | 1.6 | **0.22** | 260 | **yes** | **yes** |
| pure water | 7.8 | 5.9 | 6.0 | 0.086 | 0.014 | 654 | no | no |
| only solute effect | 13 | 5.8 | 6.0 | 0.21 | 0.035 | 654 | no | no |
| without reactivation | 18 | 7.9 | 10 | 1.1 | 0.11 | 111 | **yes** | no |
| low $N_a$ | 16 | 9.6 | 11 | 0.40 | 0.036 | 92 | no | no |
| high $N_a$ | 17 | 3.1 | 4.7 | 1.5 | **0.32** | 913 | **yes** | **yes** |
| low $w$ | 13 | 7.7 | 8.8 | 0.60 | 0.068 | 191 | **yes** | no |
| high $w$ | 17 | 4.6 | 5.3 | 1.0 | **0.19** | 695 | **yes** | **yes** |
| thin $\Delta H$ | 17 | 6.2 | 8.5 | 1.4 | **0.16** | 192 | **yes** | **yes** |
| thick $\Delta H$ | 9.0 | 4.1 | 5.8 | 0.50 | 0.087 | 654 | no | **yes** |

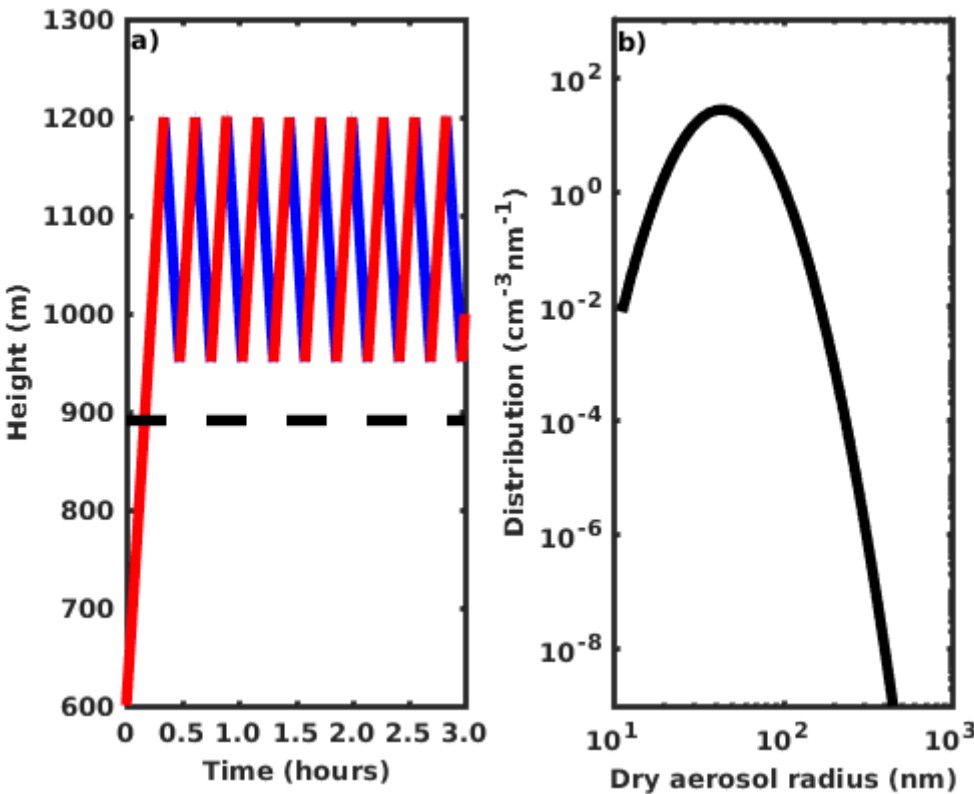

**Figure 1.** a) Trajectory of cloud parcel with upward and downward oscillations. Velocity is constant and is $0.5\ m\ s^{-1}$ for the ascending parcel and $-0.5\ m\ s^{-1}$ for the descending parcel. The dashed line is the cloud base, and the red and blue lines represent ascending and descending parcels. b) Initial dry aerosol size distribution. The total aerosol number concentration is $1000\ cm^{-3}$.

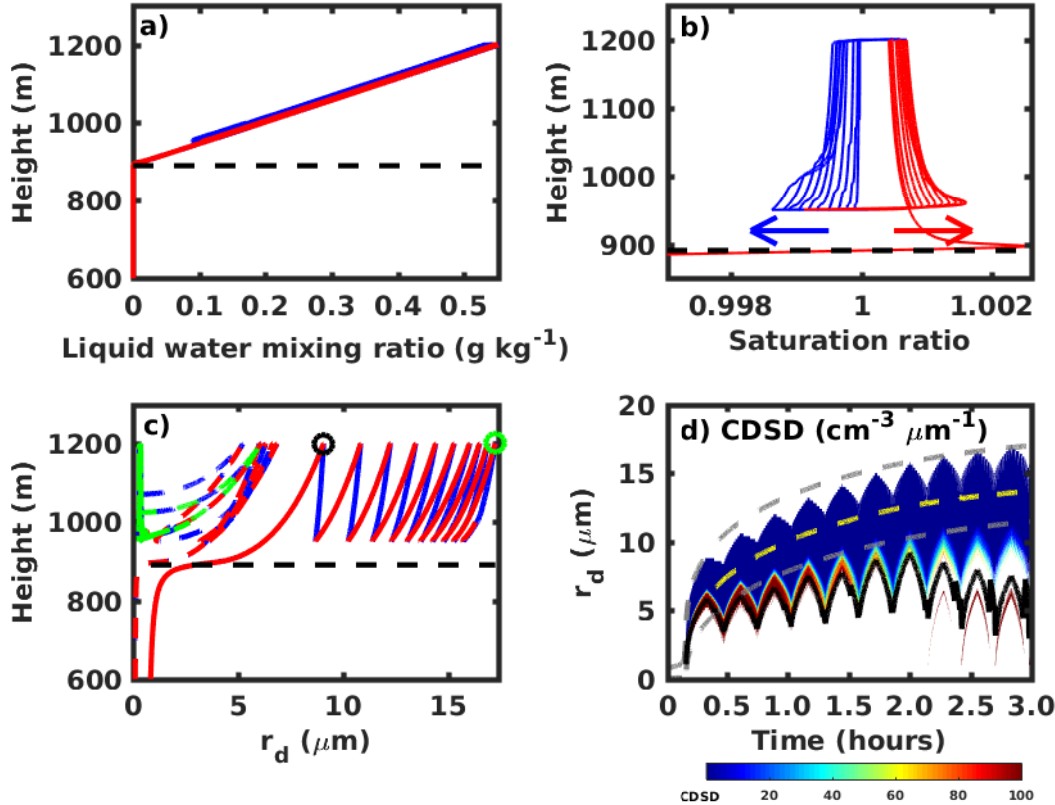

**Figure 2.** Thermodynamical and microphysical properties of an adiabatic cloud parcel with upward and downward oscillations. a) Liquid water mixing ratio changes with height. b) Cloud parcel saturation ratio changes with height. Arrows in b represent the evolution of saturation ratio profile with time. c) Radii changes of two selected cloud droplets with height. The solid line is for the largest cloud droplet that formed on a dry aerosol with radius of $503\ nm$, and the dashed line is for droplet that formed on an aerosol of $51\ nm$. The red and blue lines in a-c represent ascending and descending parcels, and the black dashed line indicates cloud base height. The green dashed line indicates the reactivation of that grid. The black and green circles are referred to in the text. d) Cloud droplet size distribution changes with time. The black line represents the mean cloud droplet radius change with time. The yellow dashed line is the change in mean droplet size for the ascending-only cloud parcel with a constant velocity of $0.5\ m\ s^{-1}$, and the upper and lower dashed gray lines represent the largest and smallest cloud droplets in the ascending-only parcel.

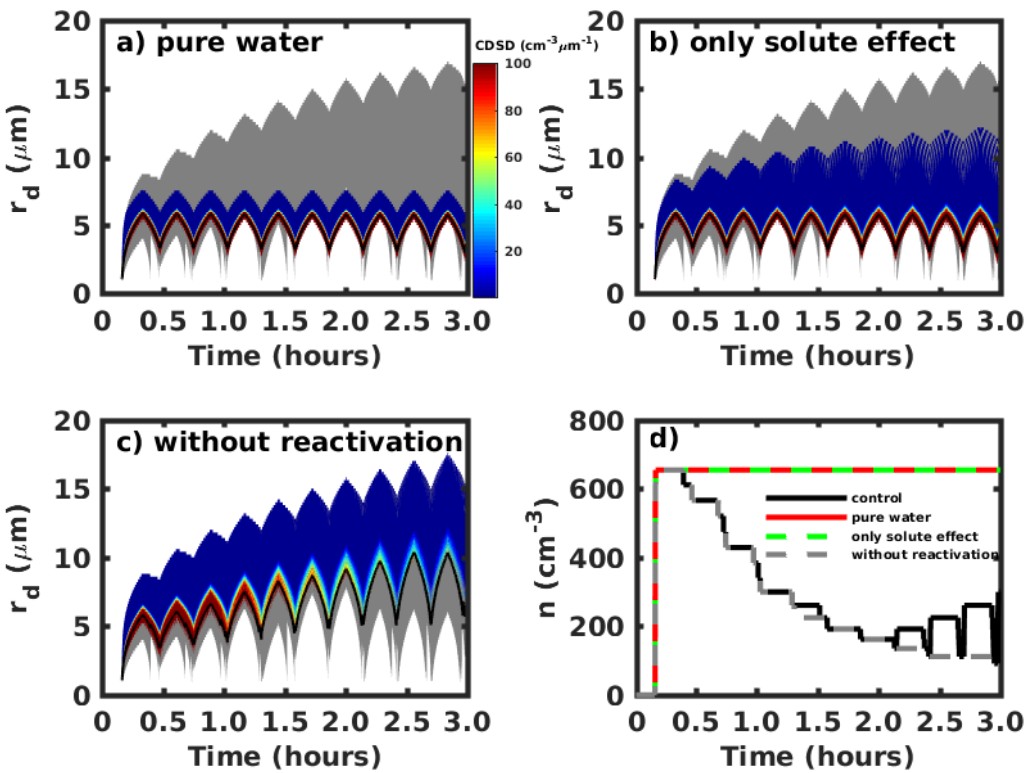

**Figure 3.** a) Cloud droplet size distribution (CDSD) changes with time without solute or curvature effects. b) CDSC changes with time with the solute effect but without the curvature effect. c) CDSD changes with time including both solute and curvature effects but where droplet reactivation is not considered. d) Total cloud droplet number concentration ($n$) changes with time for the different cases. The gray region in a-c represents the range of the droplet size spectrum for the control case, and the black lines represent the mean cloud droplet radius change with time.

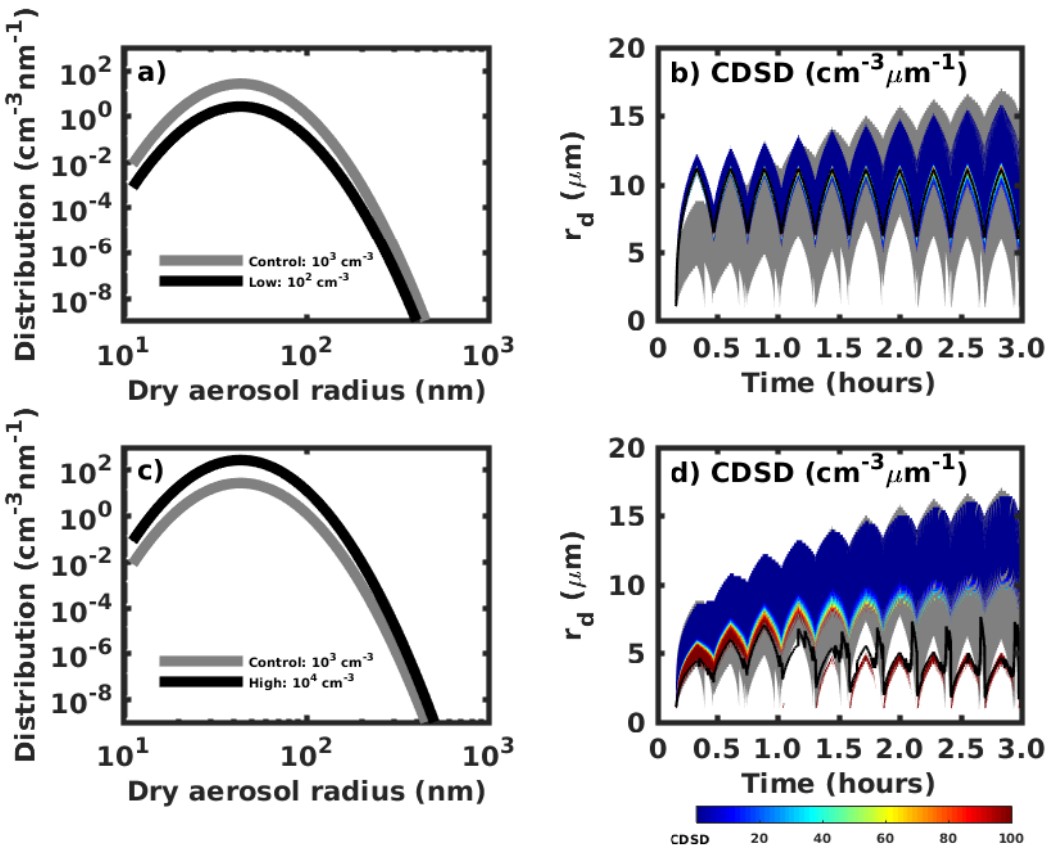

**Figure 4.** a) Aerosol size distribution for a low number concentration of $10^2$ $cm^{-3}$. b) Cloud droplet size distribution changes with time for the low aerosol number concentration case. c) Aerosol size distribution for the high number concentration of $10^4$ $cm^{-3}$. d) Cloud droplet size distribution changes with time for the high aerosol number concentration case. Gray lines in a and c represent the control case with a total aerosol number concentration of $10^3$ $cm^3$, and gray regions in b and d are the range of the cloud droplet size spectrum for the control case.

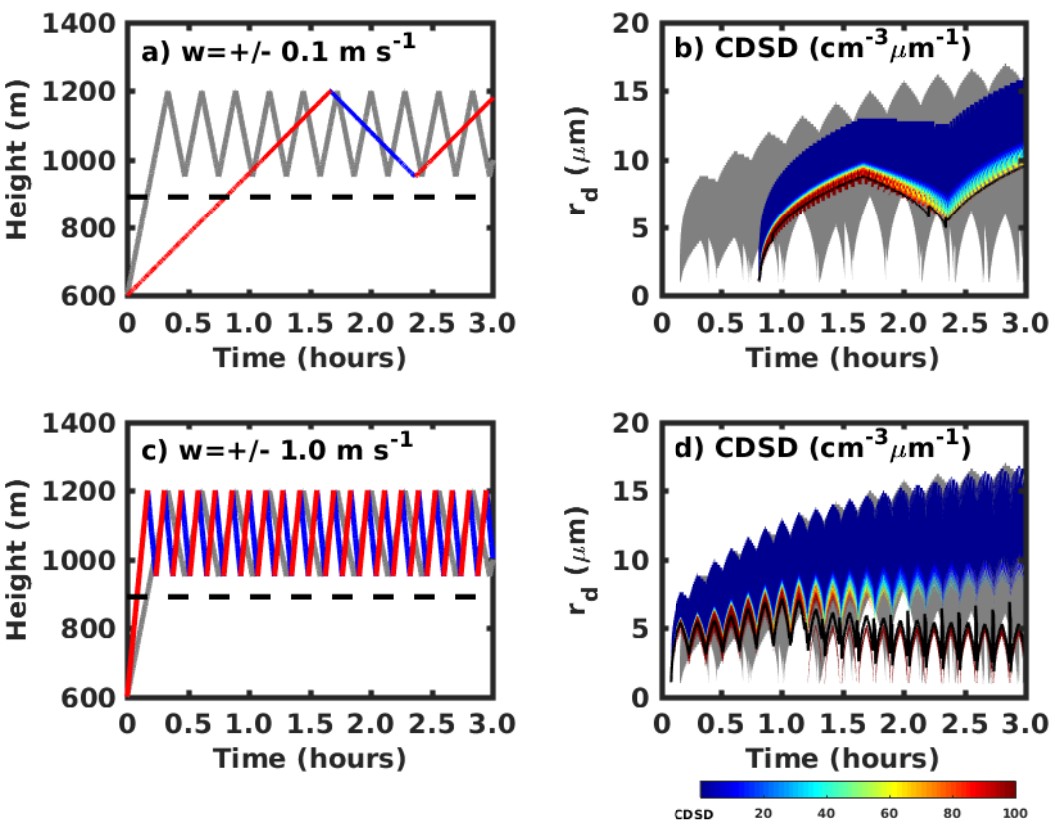

**Figure 5.** a) The height of cloud parcel changes with time for the low velocity case of $\pm\, 0.1\ ms^{-1}$. b) Cloud droplet size distribution changes with time for the low velocity case. c) The height of the cloud parcel changes with time for the velocity of $\pm\, 1.0\ ms^{-1}$. d) Cloud droplet size distribution changes with time for the high velocity case. Gray lines in a and c represent the control case with velocity of $\pm 0.5\ ms^{-1}$, and the gray regions in b and d are the range of cloud droplet spectrum for the control case.

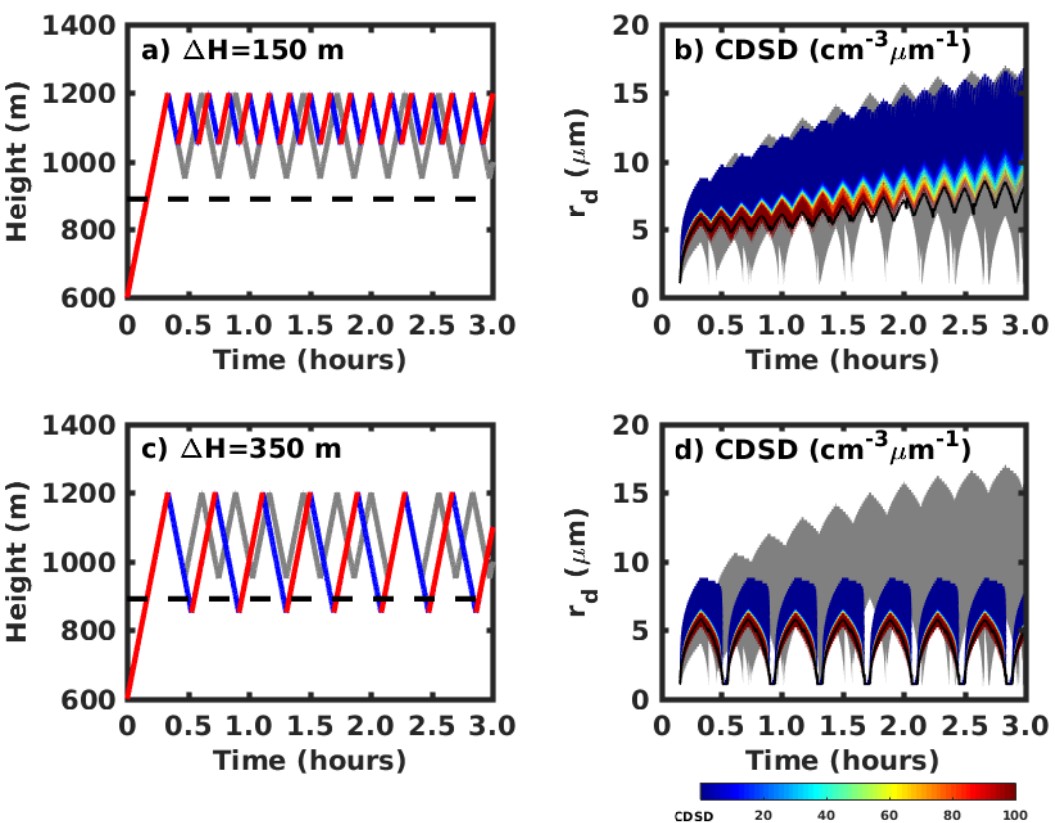

**Figure 6.** a) The height of cloud parcel changes with time for the thin recirculation layer of $150\ m$. b) Cloud droplet size distribution changes with time for the thin recirculation layer case. c) Aerosol size distribution for the thick recirculation layer of $350\ m$. d) Cloud droplet size distribution changes with time for the thick recirculation layer case. The gray lines in a and c represent the control case with recirculation layer of $250\ m$, and the gray regions in b and d are the range of cloud droplet size spectrum for the control case.

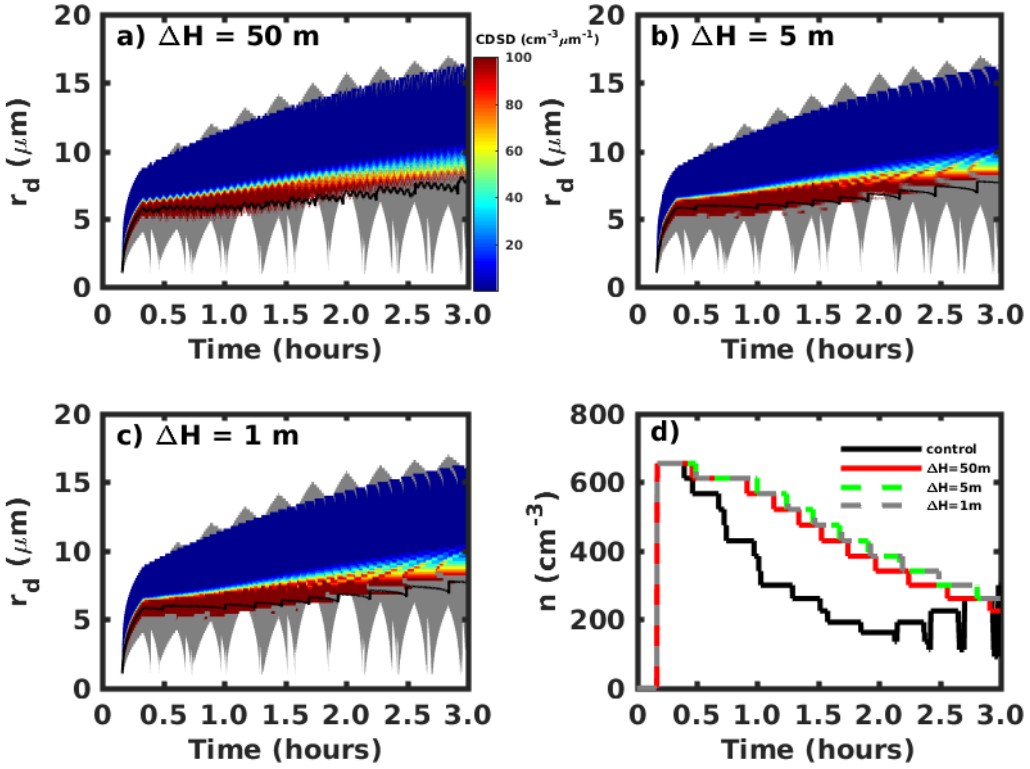

**Figure 7.** Cloud droplet size distribution (CDSD) changes with time for different thicknesses of recirculation layers: a) $\Delta H = 50\ m$, b) $\Delta H = 5\ m$, c) $\Delta H = 1\ m$. d) Total cloud droplet number concentration ($n$) changes with time for the different cases. The gray region in a-c represents the range of the droplet size spectrum for the control case, and the black lines represent the mean cloud droplet radius change with time.

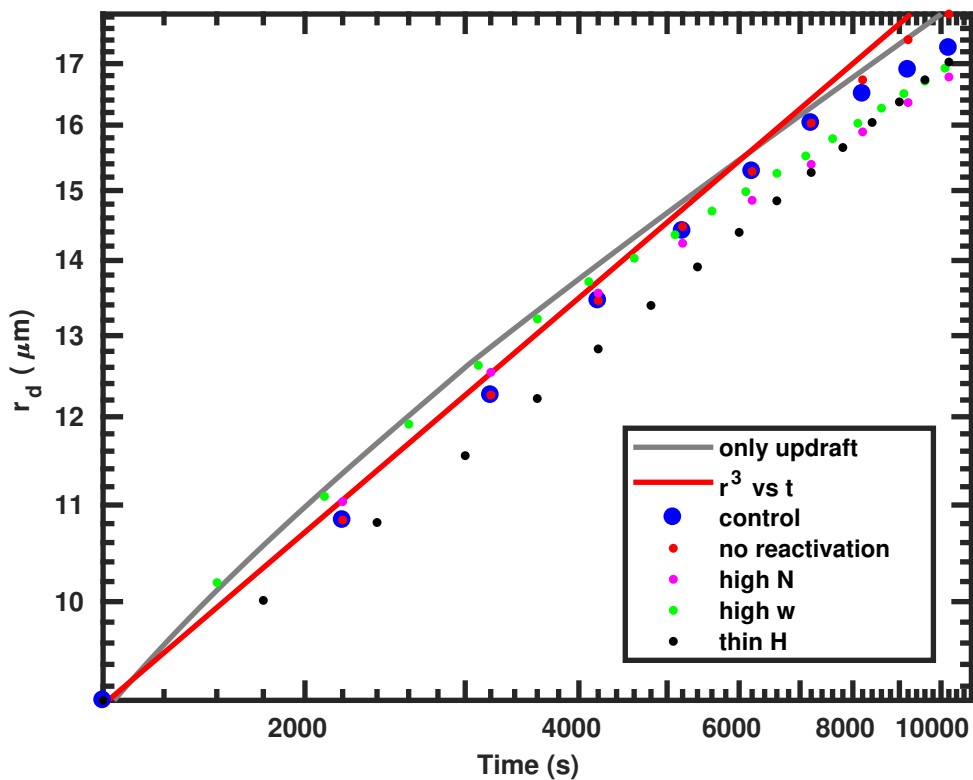

**Figure 8.** The largest cloud droplet size after each cycle is plotted for different cases discussed before: blue dots, control case; red dots, no reactivation case; pink dots, high number concentration case; green dots, high vertical velocity case; and black, thin oscillation layer case. The gray line is for the ascending-only case from Figure 4, and the red line represents the growth of a droplet.

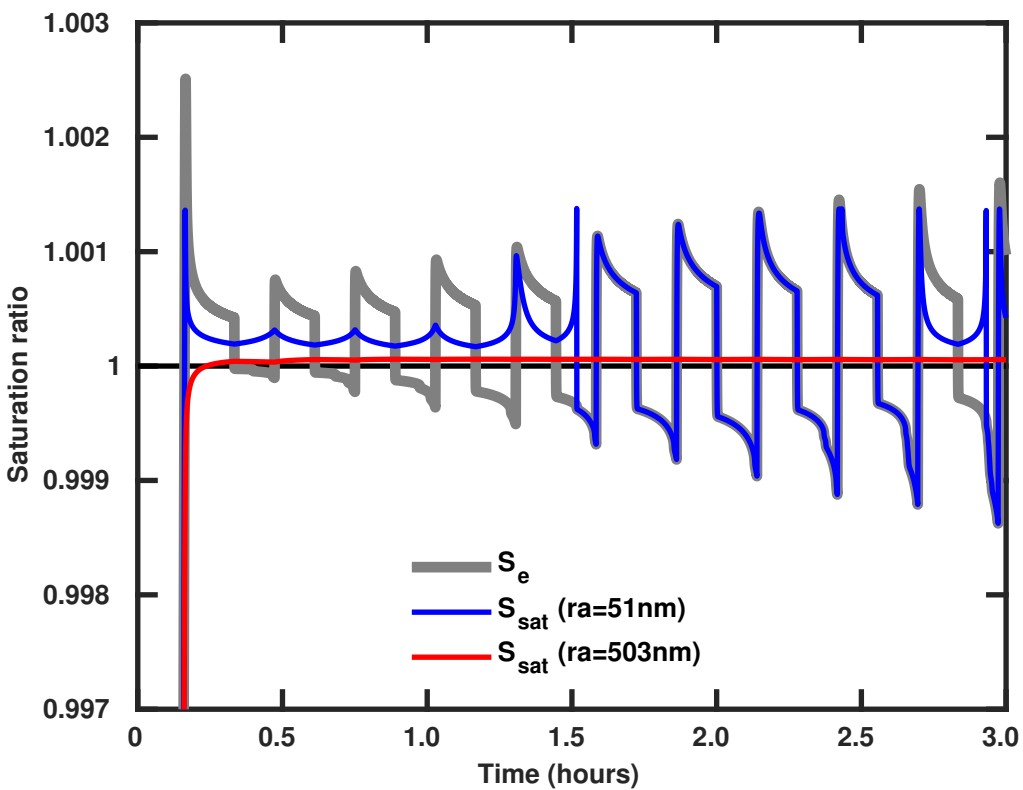

**Figure 9.** Changes of environmental saturation ratio (grey) and equilibrium saturation ratios over two droplets (red and blue) with time in an oscillating parcel. The blue line is for a droplet formed on a dry aerosol with radius of $53\ nm$ and the red line is for a droplet formed on a dry aerosol with radius of $503\ nm$. The smaller cloud droplet (formed on a dry aerosol with radius of $53\ nm$) deactivates at approximately 1.5 hours and reactivates at approximately 2.5 hours.