# Peer review of "Cloud droplet size distribution broadening during diffusional growth: ripening amplified by deactivation and reactivation"

_Atmospheric Chemistry and Physics, 2017_

## Referee Comment (RC1) · S. Arabas (Referee) · 2 Jan 2018

Comments to manuscript acpd-2017-1125:
**Cloud droplet size distribution broadening during diffusional growth:**
**ripening amplified by deactivation and reactivation**
* * *
The reviewed manuscript discusses the phenomenon of cloud droplet size spectrum broadening using an adiabatic parcel model. The authors highlight the role of the interplay between condensation/evaporation on small and large particles leading to an irreversible process analogous to Ostwald ripening. A methodology for discerning the contributions of deactivation and reactivation is developed and used to depict the amplifying role of deactivation and activation for the ripening-induced broadening. The topic is of prime relevance in the context of the ongoing developments of models comprehensively accounting for two-way aerosol-cloud interactions.

In general, the paper is concise and interesting, and I do recommend its publication pending revisions addressing concerns detailed below, and mainly related to:
- noncomprehensive presentation of earlier works on the topic,
- insufficient discussion of the limitations of the presented approach,
- limited reproducibility of the study.

**Comments on the content**

**Abstract**

The study builds upon the considerations presented by Korolev in 1995, what is dully acknowledged. However, the work of Çelik and Marwitz (1999), which is elsewhere (e.g., Wood et al. 2002) credited as the first to depict the Ostwald ripening in the context of cloud droplet growth, is not mentioned. Let me suggest not to include any references in the abstract, but rather revisit the introductory section to provide comprehensive references to earlier works on the topic including Srivastava (1991) and Çelik and Marwitz (1999).

The manuscript mentions turbulence only within the abstract and in the conclusions section (plus the somehow less relevant reference to turbulence-induced enhancement in collision efficiency on page 2). Some discussion is needed in the text to warrant statements that the study addresses turbulence-relevant vertical oscillations. In particular, the frequencies of oscillations studied are distant from those considered in recent studies on turbulence-induced effects in air-parcel activation models, e.g., (Ditas et al., 2012, Fig. 10 therein) or Hammer et al. (2015, Fig. 10 therein).

Let me also suggest using "moving-bin" instead of "Lagrangian bin-microphysics" in the abstract and throughout the text.

**Section 1**

A complete rewrite of the second paragraph (p. 2, lines 10–30) would be a good idea. The first sentence could likely be moved to the beginning of the third paragraph, perhaps made more precise by mentioning aerosol spectrum (or even moving-bin representation), and supported with some classic reference, e.g., the already referenced work of Mordy, but perhaps also the seminal work of Howell (1949). The second and third sentences could be merged in into the first paragraph where both narrow spectrum and cloud parcel are already mentioned. Then, I would suggest splitting the rest of the paragraph into two separate ones on: (i) the possible causes, and (ii) the possible effects of the broadening of cloud droplet spectrum.

Among the causes, the influence of aerosols highlighted in the already cited work of Chandrakar et al., the influence of in-cloud activation (e.g., Khain et al., 2000, sect. 3.5) as well as of turbulence (Devenish et al., 2012, e.g.,) could be mentioned additionally. The recent work of Grabowski and Abade, 2017 seems relevant to me as well.

Among the effects, along with the already mentioned enhancement of collision efficiency, the optical aspects should be listed given that they are highlighted even in the first sentence of the abstract. In fact, the last sentence of section 3.1 (p. 6, lines 12-13) seems to me to be more appropriate here.

It would be also beneficial to clarify the meaning of supersaturation fluctuations as the same term is used for studies assuming uniform supersaturation within an air parcel (as in Korolev, 1995) as well as studies resolving inhomogeneities of supersaturation in space (Devenish et al., 2012, and references therein).

The phrases "irreversibility of droplet size spectrum shape" (p. 2, line 26), "CDSD is irreversible" and "Irreversibility of the CDSD" (p. 4, line 13), while consistent with the original wording of Korolev (1995) sound somehow confusing to me as it is the process (i.e., the evolution in time) that is irreversible and not the spectrum shape – just a nomenclature issue. On a related note, the discussion on hysteretic effects in activation-deactivation cycles presented in Arabas and Shima (2017) might be of relevance (although limited to monodisperse spectra).

**Section 2**

ACP guidelines clearly state that "*paper should contain sufficient detail and references to public sources of information to permit the author's peers to replicate the work*"[1]. It is thus essential to either comprehensively define the mathematical formulation of the employed model or provide a straightforward way of obtaining the employed software in the very revision used for obtaining presented results.

To highlight the problem, let me point out that the Feingold, Walko, et al. (1998) paper referenced as describing "*the original version of the model*" actually covers simulations with a 2D LES-type model "*that uses lognormal basis functions to represent cloud and drizzle drop spectra*". The Feingold, Kreidenweis, and Zhang (1998) reference was likely meant, although therein the reader is refereed to Feingold and Heymsfield (1992) for "*further details of the microphysical model*". There, in turn, the reader will learn that "*the model used ... is discussed in detail by Heymsfield and Sabin (1989)*". While a parcel model might be considered a very simple tool, the numerical nuances (e.g., spectral discretisation, implicit vs. explicit supersaturation calculation, choice of values for parameters such as mass accommodation coefficient) do cause significant differences among results from different implementations as depicted for instance in the intercomparison study of Kreidenweis et al. (2003) which actually included the model used in the refereed manuscript. While properly attributing the authors of model formulation and implementation, and giving the readers the ability to reproduce the results is crucial, elaborating on the model details shall make the manuscript easier to comprehend as well.

**Section 3.1**

The references to sizes of single cloud droplets (p. 4, lines 2, 3, 24, 26, 30, 31; p. 8, l. 3) contrast the more appropriate description of "droplet size for a bin" (p. 3, lines 20, 21, 25; p. 4 line 4). There is also a statement on "reactivation of that bin" (caption of Fig. 2). I suggest unifying the way the size associated with a moving bin is referred to.

The notion of "totally evaporated" droplet (p. 4, line 30) seems misleading to me. The model describes a population of solution droplets, likely under the assumption of the salt mass being negligible in comparison with water mass. Conditions imposed to disable reactivation should be clarified.

I suggest rephrasing the passage on supercooled parcel at 6000 m to underline the technical (not physical) nature of this element of the analysis.
* * *
[1] http://www.atmospheric-chemistry-and-physics.net/for_authors/obligations_for_authors.html

**Section 3.2**

This section lacks any references to other studies which would be very appropriate here and which should help to give support to the choice of parameters used.

As will be pointed out below, the analysis of sensitivity to spectral discretisation would also be very beneficial.

**Section 3.3**

As a general comment, let me point out that neither the sensitivity analysis nor the discussion of the results touches upon the numerical limitations of the employed parcel model. As pointed out in Kreidenweis et al. (2003, e.g., discussion of Fig. 8 therein) both the spectral discretisation and the uncertainty in the value of mass accommodation coefficient translate into significant uncertainty in the results (obtained with the very same parcel model as used in this study). In Takeda and Kuba (1982, sect. 2.5 therein) it was pointed out that the narrowness of size distributions reported by Mordy (1959) was actually likely influenced by the spectrum discretisation. As a more technical remark, the analysis presented in Arabas and Pawlowska (2011, Fig. 4 therein) shall discourage the authors from using three-significant-digit precision in Table 1 and throughout the paper.

The last sentence of the first paragraph (p. 7, lines 24-26) shall likely be extended into a separate paragraph to allow for referencing the discussion that followed from the work of Liu and Daum – see e.g. Lu and Seinfeld (2006, sect. 6 therein) and Brenguier, Burnet, and Geoffroy (2011, sect. 2 therein). Also, the issue of instrumental broadening shall be mentioned (sect. 3.2 in Devenish et al., 2012, and references therein).

The discussion of residence time in the third paragraph (p. 8) could benefit from referencing other studies discussing in-cloud residence time in context of aerosol recycling (see e.g. section 4.2 in Andrejczuk, Reisner, et al., 2008, and references therein).

**Section 4**

The discussion on the limitations of the presented analysis given in second and third paragraph of the section (p. 9 lines 31-33, p.10 lines 1-13) is somehow imbalanced, in my opinion. On the one hand, the lack of entertainment and mixing is commented just with a short statement. On the other hand, a separate paragraph is presented in support of the assumption of polydisperse aerosol and the presence of both upward and downward motions (if to be kept, this paragraph calls for references and more quantitative discussion, e.g. by discussing the relevant dynamical and microphysical timescales as in Korolev 1995, sect. 6). I suggest placing much more attention on the adiabaticity assumption, especially given the three-hour-long simulation time. The discussion of the importance of mixing based on LES and TEM simulations presented in Ovchinnikov and Easter (2010) shall come in handy, especially that the TEM used therein is based on the same parcel-model formulation from Feingold, Kreidenweis, and Zhang (1998).

The discussion presented in the last paragraph (p. 10, lines 15-21 also referenced in the last sentence of the abstract) calls for a mention of particle-based microphysics techniques, some of which do fulfil the mentioned requirement of considering "*both solute and curvature effects before and after activation*", and in particular – also deactivation and reactivation. Several references to works published throughout the last decade are given, e.g., in Hoffmann, Raasch, and Noh (2015), where discussion on the role of reactivation can also be found (sect. 3.1 therein).

While it might likely be considered out of scope of the present paper, let me point out that the presented discussion is a very counterargument to the simplification of the particle-based condensation schemes recently suggested in Grabowski, Dziekan, and Pawlowska (2017), and based on the assumption that detailed modelling of reactivation is only relevant if aerosol processing by collisions or chemical reactions is addressed. The earlier discussions of the consequences of neglecting pre-activation droplet growth in models of clouds (e.g., Srivastava, 1991; Chuang, Charlson, and Seinfeld, 1997) seem relevant as well.

Finally, the authors shall consider citing Ovchinnikov and Easter (2010) along the work of Lebo and Seinfeld (p. 10, line 18), while the reference to the work of Bott, focused on the coalescence numerics, seems less relevant. References to earlier works employing joint "2d-bin" aerosol-cloud spectra can be found e.g. in paragraph 3 of Andrejczuk, Grabowski, et al. (2010) and in paragraph 10 of Ovchinnikov and Easter (2010).

**Comments on the composition and technical remarks**

**p. 1, l. 23** please avoid the word "believed"

**p. 1, l. 24** Imagining rather than imaging?

**p. 2, l. 2** please explain or remove the word "linear"

**p. 2, l. 3** please rephrase the sentence so that collisional growth efficiency is not logically coupled with inverse proportionality of condensational growth rate

**p. 2, l. 3-4** I suggest using approximate sizes and perhaps referencing a more recent textbook instead of the work of Hocking

**p. 2, l. 15** please indicate causation instead of just saying "be related to"

**p. 2, l. 23** GCCN provide (and not provides)?

**p. 3, l. 3** sensitivity studies (not sensitivities)

**p. 5, l. 24** "kinetic" (i.e., relate to the pace of the process as in chemical kinetics) rather than "kinematic" (i.e., related to motion)?

**p. 6, l. 20** please rephrase "number concentrations of the control"

**p. 7, l. 15** "larger than **in** the control case"

Within references, please correct capitalisation in journal names and use abbreviated versions following the ACP guidelines[2]. I strongly suggest adding a doi label for each reference (this will not be added by Copernicus editors). Here are corrections to several entries in the bibliography:

- Bott reference volume should be 59–60.
- Cheng et al. reference is missing page identifier: D08201.
- Falkovich and Pumir reference has wrong year (2015, should be 2007), wrong volume (should be 64) and is missing page numbers: 4497–4505.
- Feingold and Siebert reference is missing book title, editor and publisher information.
- Heintzenberg et al. reference has a truncated title and missing booktitle information, it should likely be replaced with Pöschl, Rose, and Andreae (2009).
- Laird et al. reference requires correction in capitalisation of "Iii".
- Li et al. reference is missing page range: 11213–11227.
- Lozar and Muessle reference should be cited as "de Lozar and Muessle" (at least according to ACP website).
- Pruppacher and Klett book reference mistakenly includes an additional author and is missing publisher name.
- Xue and Feingold reference is missing page identifier: D18204.

**Figures**

It is essential to replace the raster low-resolution image files used in figures 1–6 with vector graphics (PostScript/SVG/PDF formats).

Hope that helps!
Sylwester Arabas
* * *
[2]http://www.atmospheric-chemistry-and-physics.net/Copernicus_Publications_Reference_Types.pdf

**References (entries present in the paper omitted)**

Andrejczuk, M., W.W. Grabowski, et al. (2010). Cloud-aerosol interactions for boundary layer stratocumulus in the Lagrangian Cloud Model. *J. Geophys. Res.* 115, D22214. DOI: 10.1029/2010JD014248.

Andrejczuk, M., J.M. Reisner, et al. (2008). The potential impacts of pollution on a nondrizzling stratus deck: Does aerosol number matter more than type? *J. Geophys. Res.* 113, D19204. DOI: 10.1029/2007JD009445.

Arabas, S. and H. Pawlowska (2011). Adaptive method of lines for multi-component aerosol condensational growth and CCN activation. *Geosci. Model Dev.* 4, 15–31. DOI: 10.5194/gmd-4-15-2011.

Arabas, S. and S. Shima (2017). On the CCN (de)activation nonlinearities. *Nonlin. Proc. Geophys.* 24, 535–542. DOI: 10.5194/npg-24-535-2017.

Brenguier, J.-L., F. Burnet, and O. Geoffroy (2011). Cloud optical thickness and liquid water path - does the k coefficient vary with droplet concentration? *Atmos. Chem. Phys.* 11, 9771–9786. DOI: 10.5194/acp-11-9771-2011.

Çelik, F. and J.D. Marwitz (1999). Droplet Spectra Broadening by Ripening Process. Part I: Roles of Curvature and Salinity of Cloud Droplets. *J. Atmos. Sci.* 56, 3091–3105. DOI: 10.1175/1520-0469(1999)056<3091:DSBBRP>2.0.CO;2.

Chuang, P.Y., R.J. Charlson, and J.H. Seinfeld (1997). Kinetic limitations on droplet formation in clouds. *Nature* 390, 594–596. DOI: 10.1038/37576.

Devenish, B. J. et al. (2012). Droplet growth in warm turbulent clouds. *Q.J.R. Meteorol. Soc.* 138, 1401–1429. DOI: 10.1002/qj.1897.

Ditas, F. et al. (2012). Aerosols-cloud microphysics-thermodynamics-turbulence: evaluating supersaturation in a marine stratocumulus cloud. *Atmos. Chem. Phys.* 12, 2459–2468. DOI: 10.5194/acp-12-2459-2012.

Feingold, G. and A.J. Heymsfield (1992). Parameterizations of Condensational Growth of Droplets for Use in General Circulation Models. *J. Atmos. Sci.* 49, 2325–2342. DOI: 10.1175/1520-0469(1992)049<2325:POCGOD>2.0.CO;2.

Feingold, G., S.M. Kreidenweis, and Y. Zhang (1998). Stratocumulus processing of gases and cloud condensation nuclei, part I, Trajectory ensemble model. *J. Geophys. Res.* 103, 19527–19542. DOI: 10.1029/98JD01750.

Feingold, G., R.L. Walko, et al. (1998). Simulations of marine stratocumulus using a new microphysical parameterization scheme. *Atmos. Res.* 47–48, 505–528. DOI: 10.1.1.453.2319.

Grabowski, W.W. and G.C. Abade (2017). Broadening of Cloud Droplet Spectra through Eddy Hopping: Turbulent Adiabatic Parcel Simulations. *J. Atmos. Sci.* DOI: 10.1175/JAS-D-17-0043.1.

Grabowski, W.W., P. Dziekan, and H. Pawlowska (2017). Lagrangian condensation microphysics with Twomey CCN activation. *Geosci. Model. Dev.* accepted. DOI: 10.5194/gmd-2017-214.

Hammer, E. et al. (2015). Sensitivity estimations for cloud droplet formation in the vicinity of the high-alpine research station Jungfraujoch (3580 m a.s.l.) *Atmos. Chem. Phys.* 15, 10309–10323. DOI: 10.5194/acp-15-10309-2015.

Heymsfield, A.J. and R.M. Sabin (1989). Cirrus Crystal Nucleation by Homogeneous Freezing of Solution Droplets. *J. Atmos. Sci.* 46, 2252–2264. DOI: 10.1175/1520-0469(1989)046<2252:CCNBHF>2.0.CO;2.

Hoffmann, F., S. Raasch, and Y. Noh (2015). Entrainment of aerosols and their activation in a shallow cumulus cloud studied with a coupled LCM–LES approach. *Atmos. Res.* 156, 43–57. DOI: 10.1016/j.atmosres.2014.12.008.

Howell, W. (1949). The growth of cloud drops in uniformly cooled air. *J. Meteorol.* 6.2, 134–149. DOI: 10.1175/1520-0469(1949)006<0134:TGOCDI>2.0.CO;2.

Khain, A. et al. (2000). Notes on the state-of-the-art numerical modeling of cloud microphysics. *Atmos. Res.* 55, 159–224. DOI: 10.1016/S0169-8095(00)00064-8.

Kreidenweis, S.M. et al. (2003). Modification of aerosol mass and size distribution due to aqueous phase $SO_2$ oxidation in clouds: comparisons of several models. *J. Geophys. Res.* 108.D7, 4213. DOI: 10.1029/2002JD002697.

Lu, M.-L. and J.H. Seinfeld (2006). Effect of aerosol number concentration on cloud droplet dispersion: A large-eddy simulation study and implications for aerosol indirect forcing. *J. Geophys. Res.* 111, D02207. DOI: 10.1029/2005JD006419.

Ovchinnikov, M. and R.C. Easter (2010). Modeling aerosol growth by aqueous chemistry in a nonprecipitating stratiform cloud. *J. Geophys. Res.* 115, D14210. DOI: 10.1029/2009JD012816.

Pöschl, U., D. Rose, and M.O. Andreae (2009). Climatologies of Cloud-related Aerosols. Part 2: Particle Hygroscopicity and Cloud Condensation Nucleus Activity. In: *Clouds in the perturbed climate system: their relationship to energy balance, atmospheric dynamics, and precipitation*. Ed. by J. Heintzenberg and R.J. Charlson. MIT Press. DOI: 10.7551/mitpress/9780262012874.001.0001.

Srivastava, R.S. (1991). Growth of cloud drops by condensation: Effect of surface tension on the dispersion of drop sizes. *J. Atmos. Sci.* 48, 1596–1605. DOI: 10.1175/1520-0469(1991)048<1596:GOCDBC>2.0.CO;2.

Takeda, T. and N. Kuba (1982). Numerical study of the effect of CCN on the size distribution of cloud droplets. Part I. Cloud droplets in the stage of condensational growth. *J. Meteorol. Soc. Jpn* 60, 978–993. DOI: 10.2151/jmsj1965.61.3_375.

---

## Referee Comment (RC2) · Anonymous Referee #2 · 2 Jan 2018

**Review of "Cloud droplet size distribution broadening during diffusional growth: ripening amplified by deactivation and reactivation" by Yang et al. submitted for Atmospheric Chemistry and Physics**

Using idealized adiabatic parcel simulations with Lagrangian bin-microphysics, the authors investigate the broadening of a cloud droplet size distribution (CDSD). By including the effects of aerosol deactivation and reactivation, it is shown that process of Ostwald ripening, which has been assumed to be weak in warm clouds by other authors, can be significantly amplified, resulting in sufficiently large droplets that might be able to initiate collision and coalescence. The authors demonstrate convincingly that the deactivation of aerosols in a downdraft leads to a lower number of cloud droplets in a subsequent updraft, which enhances the growth of these droplets, resulting in superadiabatic droplets sizes. Additionally, the reactivation of some aerosols leads to an additional broadening of the CDSD to smaller sizes.

Although I feel that the presented results represent a rather extreme case of the amplification of Ostwald ripening due to aerosol deactivation/reactivation, which might not be the case in nature, it clearly demonstrates the effect and potential importance of a proper representation of deactivation/reactivation, which many cloud models lack. Accordingly, some minor additional simulations might be necessary to determine the limits of the presented microphysical processes and to fit it in the current literature. All in all, the manuscript is interesting, well written, and should be published after the following concerns are addressed.

**General Comments**

*Model Description.* Although plenty of references are given, the essential parts of the used microphysical model need to be stated. Only the abstract and the conclusions (Sec. 4) state, that the bin-microphysics is Lagrangian, i.e., it utilizes moving bins instead of fixed bins. This information is missing in Sec. 2 but essential for the model used in this study, which relies on a fixed relation of aerosol mass and droplet size (which is only possible in a Lagrangian (or moving bin) framework). Does the microphysical model include any other processes than diffusional droplet growth including activation/deactivation? Moreover, it would be nice (but not necessary) to present the used equation describing diffusional droplet growth including activation/deactivation. This would be also an opportunity to define quantities as $S_e$ and $S_{sat}$, which are used in other parts of the manuscript (e.g., Fig. 8).

*Idealized Setup.* It is not disputable that the presented simulations represent an idealized setup. However, the probability that a parcel undergoes numerous oscillations of 150 m or more is rather unlikely. The results of Wood et al. (2002), who investigated CDSD ripening in a slightly more realistic setup including potential effects of aerosol deactivation and reactivation (last lines of their section 3), do not indicate a strong evidence of the proposed amplification of CDSD ripening by deactivation/reactivation. Therefore, I strongly suggest testing even thinner recirculation layers, i.e., fluctuations which are more likely to be observed in nature. I expect that if a certain depth of the recirculation layer is undercut, deactivation will be inhibited and the amplification of ripening due to deactivation/reactivation will stop. These additional investigations are not only necessary to understand the importance of the proposed amplification mechanism, but also connects the presented study to other work on spectral ripening (e.g., Wood et al. (2002), or Grabowski and Abade (2017) who extensively investigated the dependence of spectral broadening on the length scales of the involved turbulence in the absence of deactivation/reactivation).

**Minor Comments**

P. 2, l. 2: Does the "linear growth rate" refer to the temporal change of the radius ($dr/dt = …$)?

P. 3, ll. 30 – 31: Give a short explanation why the liquid water is slightly smaller in the ascending branch compared to the descending.

P. 4, ll. 7 – 8: Although it has been stated before, I would mention the development of a second mode in the CDSD by the reactivation of aerosols after about 2 hours.

Fig. 2d: How do the aerosol masses (or dry radii) distribute across the CDSD? I expect that the largest droplets have been grown from the largest aerosols.

P. 5, ll. 34 – 35: Give some more explanations on the setup of the ascending-only parcel simulation. What is its vertical velocity? The answer can be deduced from the following text but a clear statement would be helpful.

P. 6, l. 17 and p. 7, l. 12: "Recycling layer"? Based on the available literature, I would prefer the name "recirculation layer".

P. 6., ll. 27 – 30: Although I agree with the interpretation that deactivation/reactivation might amplify the ripening process, an additional explanation, originating directly from Korolev (1995, Section 2), needs to be considered: Broadening only occurs if the supersaturation is smaller than maximum of $S^+(r)$, a quantity which indicates the narrowing or broadening of the spectrum in the vicinity of a certain radius $r$. If the supersaturations are generally higher than $S^+$, only narrowing of the CDSD occurs. Since in-cloud supersaturations generally decrease due to an increase in aerosol number concentration, it is to expect that only the more aerosol-laden simulations will be affected by Ostwald ripening while the cleaner simulation might be less affected (or not affected at all), which also agrees with the presented study.

Fig. 5b: Where do the high-frequent oscillations in the CDSD come from?

P. 8, l. 9: For clarity, state the underlying equation used for calculating $S_{sat}$.

P. 8, l. 24 – 34: This is an interesting result. Although I can imagine where the equation in l. 27 comes from, an extra step for its deviations might be illuminating for all readers. Moreover, I suggest discussing the underlying physics of the term $s_k$ in slightly more depth. A nice explanation is given for the case of a negative vertical velocity, in which the evaporation of a large number of small droplets maintains the supersaturation at a certain level. But how does $s_k$ act in an updraft?

P. 9, ll. 24 – 25: What is meant by the right upper boundary of the CDSD?

P. 10, ll. 15 – 21: There are models with a similar treatment of microphysics, so-called Lagrangian cloud model. And a couple of publications investigation aerosol activation/deactivation in that framework (e.g., Andrejczuk et al. 2008; Hoffmann et al., 2015; Hoffmann 2017).

**Technical Comments**

P. 1, l. 6: Usually, an abstract does not contain any citations.

P. 1, l. 20: "of a warm cloud" or "of warm clouds"

P. 2, l. 23: "… GCCN not only **provide** an embryo … but also **enhance** droplet growth …"

P. 3, l. 17: Since American English is used throughout the manuscript: "sulfate"

**References**

Andrejczuk, M., Reisner, J.M., Henson, B., Dubey, M.K. and Jeffery, C.A., 2008. The potential impacts of pollution on a nondrizzling stratus deck: Does aerosol number matter more than type?. Journal of Geophysical Research: Atmospheres, 113(D19).

Grabowski, W.W. and Abade, G.C., 2017. Broadening of Cloud Droplet Spectra through Eddy Hopping: Turbulent Adiabatic Parcel Simulations. Journal of the Atmospheric Sciences, 74(5), pp.1485-1493.

Hoffmann, F., Raasch, S. and Noh, Y., 2015. Entrainment of aerosols and their activation in a shallow cumulus cloud studied with a coupled LCM–LES approach. Atmospheric Research, 156, pp.43-57.

Hoffmann, F., 2017. On the limits of Köhler activation theory: how do collision and coalescence affect the activation of aerosols? Atmospheric Chemistry and Physics, 17(13), pp.8343-8356.

Korolev, A.V., 1995. The influence of supersaturation fluctuations on droplet size spectra formation. Journal of the Atmospheric Sciences, 52(20), pp.3620-3634.

Wood, R., Irons, S. and Jonas, P.R., 2002. How important is the spectral ripening effect in stratiform boundary layer clouds? Studies using simple trajectory analysis. Journal of the Atmospheric Sciences, 59(18), pp.2681-2693.

---

## Author Comment (AC1) · 15 Mar 2018

**Responses to reviewer 1**

Comments to manuscript acpd-2017-1125: Cloud droplet size distribution broadening during diffusional growth: ripening amplified by deactivation and reactivation The reviewed manuscript discusses the phenomenon of cloud droplet size spectrum broadening using an adiabatic parcel model. The authors highlight the role of the interplay between condensation/evaporation on small and large particles leading to an irreversible process analogous to Ostwald ripening. A methodology for discerning the contributions of deactivation and reactivation is developed and used to depict the amplifying role of deactivation and activation for the ripening-induced broadening. The topic is of prime relevance in the context of the ongoing developments of models comprehensively accounting for two-way aerosol-cloud interactions. In general, the paper is concise and interesting, and I do recommend its publication pending revisions addressing concerns detailed below, and mainly related to:

• noncomprehensive presentation of earlier works on the topic,

• insufficient discussion of the limitations of the presented approach,

• limited reproducibility of the study.

We thank the reviewer for the constructive comments and for the improvements they have motivated in the revised paper. In this document we address all comments and detail the changes. Reviewer comments are in blue, our response is in black and modifications of the manuscript are summarized in red text.

To sum up, we add several paragraphs in section 1 to give a "comprehensive presentation of earlier works on this topic". To allow the "reproducibility of this study", we add more details of our current model in section 2, including key mathematical equations. More sensitivity studies, including the effects of water accommodation coefficient and spectral discretization, and more discussion of the limitations of the presented approach are added in sections 3 and 4.

Comments on the content

Abstract

The study builds upon the considerations presented by Korolev in 1995, what is dully acknowledged. However, the work of Celik and Marwitz (1999), which is elsewhere (e.g., Wood et al. 2002) credited as the first to depict the Ostwald ripening in the context of cloud droplet growth, is not mentioned. Let me suggest not to include any references in the abstract, but rather revisit the introductory section to provide comprehensive references to earlier works on the topic including Srivastava (1991) and Celik and Marwitz (1999). The manuscript mentions turbulence only within the abstract and in the conclusions section (plus the somehow less relevant reference to turbulence-induced enhancement in collision efficiency on page 2). Some discussion is needed in the text to warrant statements that the study addresses turbulence-relevant vertical oscillations. In particular, the frequencies of oscillations studied are distant from those considered in recent

studies on turbulence-induced effects in air-parcel activation models, e.g., (Ditas et al., 2012, Fig. 10 therein) or Hammer et al. (2015, Fig. 10 therein).

Thank the reviewer for the helpful comment. We remove the reference in the abstract.

"Results show that the CDSD can be broadened during condensational growth as a result of Ostwald ripening amplified by droplet deactivation and reactivation, which is consistent with earlier work, which is consistent with Korolev (1995)."

In the introduction, we add a paragraph to have an in-depth discussion of previous studies about the Ostwald ripening effect on cloud droplet size distribution broadening.

"Ostwald ripening for cloud droplets is the phenomenon when larger droplets grow and smaller droplets shrink due curvature and/or solute effects and, thus, it can broaden the CDSD at both small and large sides of the distribution. Srivastava (1991) investigated the growth of cloud droplets in a rising air parcel. Results show that the variance of squared radius of the CDSD was constant during the condensational growth process if both curvature and solute effects were ignored, but it was increased if those effects were considered. This "condensational broadening" is more pronounced in clouds with high cloud droplet number concentration and low vertical velocity. In turbulent clouds, droplets will experience supersaturated/subsaturated conditions in updraft/downdraft regions. Korolev (1995) studied the evolution of the CDSD driven by supersaturation fluctuations in a vertically oscillating air parcel. Supersaturation fluctuations in his study mean that air is supersaturated in the updraft and subsaturated in the downdraft; however no spatial inhomogeneity of supersaturation is considered in the parcel. Results show that the growth and evaporation cycles during the CDSD evolution are irreversible if the solute and curvature effects are considered. This "CDSD irreversibility" (terminology used in his paper) will promote the growth of large cloud droplets, lead to evaporation or even deactivation of small cloud droplets, and thus broaden the CDSD. Korolev (1995) argued that stronger turbulent fluctuations of supersaturation would result in a broader CDSD. This is contrary to Celik and Marwitz (1999), who found that supersaturation fluctuations are not responsible for CDSD broadening and the formation of large droplets. The curvature and solute effects on Ostwald ripening, activation and deactivation have been the topics of study in recent years (e.g., Wood et al., 2002; Arabas and Shima, 2017, Chen et al., 2018; Sardina et al., 2018) but, to our knowledge, the relative roles of the curvature effect and solute effect on CDSD broadening have not been investigated."

Let me also suggest using "moving-bin" instead of "Lagrangian bin-microphysics" in the abstract and throughout the text.

We changed "Lagrangian bin-microphysics" to "moving-bin" in the abstract and throughout the manuscript. "Lagrangian bin-microphysics" is included as a parenthetical when introduced in the methods.

A complete rewrite of the second paragraph (p. 2, lines 10–30) would be a good idea. The first sentence could likely be moved to the beginning of the third paragraph, perhaps made more precise by mentioning aerosol spectrum (or even moving-bin representation), and supported with some classic reference, e.g., the already referenced work of Mordy, but perhaps also the seminal work of Howell (1949). The second and third sentences could be merged in into the first paragraph where both narrow spectrum and cloud parcel are already mentioned. Then, I would suggest splitting the rest of the paragraph into two separate ones on: (i) the possible causes, and (ii) the possible effects of the broadening of cloud droplet spectrum.

Thank you for the helpful comments. We split the paragraph into three: (1) the effects of the broadening of cloud droplet spectrum; (2) turbulence-induced CDSD broadening; (3) aerosol-induced CDSD broadening.

**Paragraph about the effects of the CDSD broadening:**

[revised manuscript text omitted]

Among the causes, the influence of aerosols highlighted in the already cited work of Chandrakar et al., the influence of in-cloud activation (e.g., Khain et al., 2000, sect. 3.5) as well as of turbulence (Devenish et al., 2012, e.g.,) could be mentioned additionally. The recent work of Grabowski and Abade, 2017 seems relevant to me as well.

We add more discussion about the influence of turbulence and in-cloud activation.

Among the effects, along with the already mentioned enhancement of collision efficiency, the optical aspects should be listed given that they are highlighted even in the first sentence of the abstract. In fact, the last sentence of section 3.1 (p. 6, lines 12-13) seems to me to be more appropriate here.

We add a paragraph about the effects of the broadening of cloud droplet spectrum (second paragraph in the manuscript).

It would be also beneficial to clarify the meaning of supersaturation fluctuations as the same term is used for studies assuming uniform supersaturation within an air parcel (as in Korolev, 1995) as well as studies resolving inhomogeneities of supersaturation in space (Devenish et al., 2012, and references therein).

We clarify the meaning of supersaturation fluctuations in the text:

"Korolev (1995) studied the evolution of the CDSD driven by supersaturation fluctuations in a vertically oscillating air parcel. Supersaturation fluctuations in his study mean that air is supersaturated in the updraft and subsaturated in the downdraft; however no spatial inhomogeneity of supersaturation is considered in the parcel. "

The phrases "irreversibility of droplet size spectrum shape" (p. 2, line 26), "CDSD is irreversible" and "Irreversibility of the CDSD" (p. 4, line 13), while consistent with the original wording of Korolev (1995) sound somehow confusing to me as it is the process (i.e., the evolution in time) that is irreversible and not the spectrum shape – just a nomenclature issue. On a related note, the discussion on hysteretic effects in activation-deactivation cycles presented in Arabas and Shima (2017) might be of relevance (although limited to monodisperse spectra).

We rephrase our discussion about "irreversibility" and make the statement clearer.

"Results show that the growth and evaporation cycles during the CDSD evolution are irreversible if solute and curvature effects are considered. This "CDSD irreversibility" (terminology used in his paper) will promote the growth of large cloud droplets, lead to evaporation or even deactivation of small cloud droplets, and thus broaden the CDSD."

Section 2

ACP guidelines clearly state that "paper should contain sufficient detail and references to public sources of information to permit the author's peers to replicate the work". It is thus essential to either comprehensively define the mathematical formulation of the employed model or provide a straightforward way of obtaining the employed software in the very revision used for obtaining presented results.

To highlight the problem, let me point out that the Feingold, Walko, et al. (1998) paper referenced as describing "the original version of the model" actually covers simulations with a 2D LES-type model "that uses lognormal basis functions to represent cloud and drizzle drop spectra". The Feingold, Kreidenweis, and Zhang (1998) reference was likely meant, although therein the reader is refereed to Feingold and Heymsfield (1992) for "further details of the microphysical model". There, in turn, the reader will learn that "the model used ... is discussed in detail by Heymsfield and Sabin (1989)". While a parcel model might be considered a very simple tool, the numerical nuances (e.g., spectral discretisation, implicit vs. explicit supersaturation calculation, choice of values for parameters such as mass accommodation coefficient) do cause significant differences among results from different implementations as depicted for instance in the intercomparison study of Kreidenweis et al. (2003) which actually included the model used in the refereed manuscript. While properly attributing the authors of model formulation and implementation, and giving the readers the ability to reproduce the results is crucial, elaborating on the model details shall make the manuscript easier to comprehend as well.

We thank the reviewer for helpful comments. Section 2 is rewritten with more details and mathematical equations employed in the model are provided. Please see the revised manuscript for more details.

Section 3.1

The references to sizes of single cloud droplets (p. 4, lines 2, 3, 24, 26, 30, 31; p. 8, l. 3) contrast the more appropriate description of "droplet size for a bin" (p. 3, lines 20, 21, 25; p. 4 line 4). There is also a statement on "reactivation of that bin" (caption of Fig. 2). I suggest unifying the way the size associated with a moving bin is referred to.

We change "the sizes of cloud droplets" to the more appropriate description "droplet size for a moving bin" throughout the manuscript.

The notion of "totally evaporated" droplet (p. 4, line 30) seems misleading to me. The model describes a population of solution droplets, likely under the assumption of the salt mass being negligible in comparison with water mass. Conditions imposed to disable reactivation should be clarified.

We change "totally evaporated" to "deactivates and becomes a haze particle" throughout the manuscript.

I suggest rephrasing the passage on supercooled parcel at 6000 m to underline the technical (not physical) nature of this element of the analysis.

We modify this sentence as:

"…For the latter case, the cloud parcel ascends at a vertical velocity of 0.5 ms$^{-1}$ for three hours with the same initial condition as the control case. At the end of the simulation, the cloud parcel reaches about 6000 m and cloud droplets are supercooled (around 248 K), but we ignore ice nucleation in this study."

**Section 3.2**

This section lacks any references to other studies which would be very appropriate here and which should help to give support to the choice of parameters used. As will be pointed out below, the analysis of sensitivity to spectral discretisation would also be very beneficial.

We thank the reviewer for the helpful comments. We add more discussion and references about the choice of the parameters as follows.

**For aerosol number concentration:**

"We test two other aerosol number concentrations , $10^2$ cm$^{-3}$ and $10^4$ cm$^{-3}$, and keep the median radius and geometric standard deviation the same as the control case (see Figures 4 a and c ). These values are chosen to represent the conditions for clean clouds ($10^2$ cm$^{-3}$) and polluted clouds ($10^4$ cm$^{-3}$), which are consistent with previous studies (Xue and Feingold, 2004; Chen et al., 2018)."

**For updraft velocity:**

" Two vertical velocities (0.1 ms$^{-1}$ and 1.0 ms$^{-1}$) are used to test their effects on CDSD broadening. These values are chosen based on observations that updraft in stratocumulus clouds is on the order of 0.1 ms$^{-1}$ and in cumulus clouds is on the order of 1.0 ms$^{-1}$ (Ditas et al., 2012; Katzwinkel et al., 2014)."

**For the recirculation layer:**

"Turbulence driven by cloud-top radiative cooling can result in various eddy sizes in the stratocumulus-topped boundary layer (Wood, 2012). Two different  recirculation layer depths are  tested, 150 m and 350 m, to investigate the effect of eddy size on CDSD broadening."

**Section 3.3**

As a general comment, let me point out that neither the sensitivity analysis nor the discussion of the results touches upon the numerical limitations of the employed parcel model. As pointed out

in Kreidenweis et al. (2003, e.g., discussion of Fig. 8 therein) both the spectral discretisation and the uncertainty in the value of mass accommodation coefficient translate into significant uncertainty in the results (obtained with the very same parcel model as used in this study). In Takeda and Kuba (1982, sect. 2.5 therein) it was pointed out that the narrowness of size distributions reported by Mordy (1959) was actually likely influenced by the spectrum discretisation. As a more technical remark, the analysis presented in Arabas and Pawlowska (2011, Fig. 4 therein) shall discourage the authors from using three-significant-digit precision in Table 1 and throughout the paper.

We thank the reviewer for the helpful comments. To test the effects of mass accommodation coefficient and spectrum discretization, two more sensitivity studies are added. One case sets the mass accommodation coefficient to 0.06 based on Shaw and Lamb (1999). The other case changes the number of bins from 100 to 200. Both cases show similar results compared with the control case. We also change the three-significant-digit precision to two-significant-digit precision in Table 1 and throughout the paper. We modify the text as:

"We have studied the effects of total aerosol number concentration, updraft velocity, and thickness of the recirculation layer on CDSD broadening. However we note that there are other parameters used in this study that can lead to the uncertainties in the results. For example, Takeda and Kuba (1982) found that using an insufficient number of model bins will lead to the narrow CDSD reported by Mordy (1959). Kreidenweis et al. (2003) found that both the spectral discretisation and the uncertainty in the value of mass accommodation coefficient can lead to uncertainty in the results. To test the effects of mass accommodation coefficient and spectrum discretization on the CDSD, two more sensitivity studies are conducted. One case is to set mass accommodation coefficient ($\alpha_m$) to 0.06 based on Shaw and Lamb (1999). It is expected that a smaller value of $\alpha_m$ might suppress the growth of cloud droplets. The other case is to change the number of bins from 100 to 200, while keeping other parameters the same as in the control case."

"… If reactivation also occurs, the smallest cloud droplet radius associated with a moving bin rmin is around 5 µm and the relative dispersion is larger than 0.1. It is interesting to note that low mass accommodation has a negligible effect on $r_{max}$, but it has a stronger impact on $r_{min}$. This will result in a broader CDSD compared with the control case. In addition, results for 200 bins are similar to that for the control case, which means that the 100 bins used in this study are enough to limit the uncertainty due to spectrum discretization…"

The last sentence of the first paragraph (p. 7, lines 24-26) shall likely be extended into a separate paragraph to allow for referencing the discussion that followed from the work of Liu and Daum – see e.g. Lu and Seinfeld (2006, sect. 6 therein) and Brenguier, Burnet, and Geoffroy (2011, sect. 2 therein). Also, the issue of instrumental broadening shall be mentioned (sect. 3.2 in Devenish et al., 2012, and references therein).

We extend the discussion of the observed cloud droplet size distribution and relative dispersion in the manuscript.

"Results from sensitivity studies show that relative dispersion is larger than 1.5 when both deactivation and reactivation occur (see Table 1), which is consistent with the values from observations and simulations (e.g., Miles et al., 2000; Liu and Daum, 2002; Lu and Seinfeld, 2006; Chandrakar et al., 2016). It should be mentioned that the CDSD observed in previous studies might have the problem of instrumental broadening due to low instrument resolution or long-distance averaging of the sampling volume (Brenguier et al., 2011; Devenish et al., 2012). A broad CDSD is also observed by recent holographic measurements, which limit the effect of instrument broadening and have much higher temporal and spatial resolution than other instruments, such as particle-counting probes (Beals et al., 2015; Glienke et al., 2017; Desai et al., 2018)."

The discussion of residence time in the third paragraph (p. 8) could benefit from referencing other studies discussing in-cloud residence time in context of aerosol recycling (see e.g. section 4.2 in Andrejczuk, Reisner, et al., 2008, and references therein).

We add more discussion about residence time of cloud droplets and its impact on droplet size.

"…Previous studies show that although the mean lifetime of cloud droplets is usually less than half an hour, the residence time for some lucky cloud droplets can be longer than one hour (e.g., Feingold et al., 1996; Kogan, 2006; Andrejczuk et al., 2008). Those long-lifetime cloud droplets might contribute to large droplets in the cloud, similar to long-lifetime ice particles in mixed-phase clouds (Yang et al., 2015)."

Section 4

The discussion on the limitations of the presented analysis given in second and third paragraph of the section (p. 9 lines 31-33, p.10 lines 1-13) is somehow imbalanced, in my opinion. On the one hand, the lack of entertainment and mixing is commented just with a short statement. On the other hand, a separate paragraph is presented in support of the assumption of polydisperse aerosol and the presence of both upward and downward motions (if to be kept, this paragraph calls for references and more quantitative discussion, e.g. by discussing the relevant dynamical and microphysical timescales as in Korolev 1995, sect. 6). I suggest placing much more attention on the adiabaticity assumption, especially given the three-hour-long simulation time. The discussion of the importance of mixing based on LES and TEM simulations presented in Ovchinnikov and Easter (2010) shall come in handy, especially that the TEM used therein is based on the same parcel-model formulation from Feingold, Kreidenweis, and Zhang (1998).

We add more discussion about the effect of entrainment and mixing and adiabatic assumption.

"It should be mentioned that one limitation of this study arises from the use of the adiabatic assumption for three-hour simulations. Turbulence can result in not only upward and downward

oscillations but also in entrainment and mixing (Shaw, 2003; Devenish et al., 2012). The latter can cause cloud droplet evaporation, deactivation and reactivation (Korolev et al., 2013; Yang et al., 2016). In addition, the lifetime of the cloud parcel is usually less than one hour (Andrejczuk et al., 2008). Therefore, one should be aware that results in this study are based on a very idealized state. More realistic studies should consider mixing processes where for example a trajectory ensemble model would be a suitable tool (Ovchinnikov and Easter, 2010; Feingold et al., 1998).…"

The discussion presented in the last paragraph (p. 10, lines 15-21 also referenced in the last sentence of the abstract) calls for a mention of particle-based microphysics techniques, some of which do fulfil the mentioned requirement of considering "both solute and curvature effects before and after activation", and in particular – also deactivation and reactivation. Several references to works published throughout the last decade are given, e.g., in Hoffmann, Raasch, and Noh (2015), where discussion on the role of reactivation can also be found (sect. 3.1 therein). While it might likely be considered out of scope of the present paper, let me point out that the presented discussion is a very counterargument to the simplification of the particle based condensation schemes recently suggested in Grabowski, Dziekan, and Pawlowska (2017), and based on the assumption that detailed modelling of reactivation is only relevant if aerosol processing by collisions or chemical reactions is addressed. The earlier discussions of the consequences of neglecting pre-activation droplet growth in models of clouds (e.g., Srivastava, 1991; Chuang, Charlson, and Seinfeld, 1997) seem relevant as well.

We thank the reviewer for pointing out that our results contradict several previous studies. We add more discussion in the manuscript:

"…The mechanism of CDSD broadening in this study requires the model to consider both solute and curvature effects all the time (i.e., before and after activation, deactivation and reactivation). Our results suggest the importance of solute and curvature effects to the deactivation and reactivation processes, which are consistent with previous studies (e.g., Andrejczuk et al., 2008; Hoffmann et al., 2015; Hoffmann, 2017; Chen et al., 2018). However the results are counter to some other studies where details of activation and deactivation are argued to be unimportant in the cloud simulation (e.g., Srivastava, 1991; Chuang et al., 1997; Grabowski et al., 2018).…"

Finally, the authors shall consider citing Ovchinnikov and Easter (2010) along the work of Lebo and Seinfeld (p. 10, line 18), while the reference to the work of Bott, focused on the coalescence numerics, seems less relevant. References to earlier works employing joint "2d-bin" aerosol-cloud spectra can be found e.g. in paragraph 3 of Andrejczuk, Grabowski, et al. (2010) and in paragraph 10 of Ovchinnikov and Easter (2010).

We add more discussion about 2d-bin aerosol-cloud spectra scheme in the manuscript,

"…Tracking the solute distribution for each bin of cloud droplet is possible using a joint 2-D bin aerosol-cloud microphysical scheme, but it is very computationally expensive (e.g.,  Andrejczuk et al., 2010; Ovchinnikov and Easter, 2010; Lebo and Seinfeld, 2011)…."

Comments on the composition and technical remarks

p. 1, l. 23 please avoid the word "believed"

We change "believed" to "considered".

p. 1, l. 24 Imagining rather than imaging?

We change "Imaging" to "Imagining".

p. 2, l. 2 please explain or remove the word "linear"

We remove "linear" throughout the manuscript.

p. 2, l. 3 please rephrase the sentence so that collisional growth efficiency is not logically coupled with inverse proportionality of condensational growth rate

We rephrase the sentence:

"…. On the other hand, collisional growth is efficient when the droplet diameter is larger than 38 μm (Pruppacher and Klett, 2010)."

p. 2, l. 3-4 I suggest using approximate sizes and perhaps referencing a more recent textbook instead of the work of Hocking

We change "Hocking" to "Pruppacher and Klett".

p. 2, l. 15 please indicate causation instead of just saying "be related to"

We change "be related to" to "cause"

p. 2, l. 23 GCCN provide (and not provides)?

We modify the sentence as:

"They found that GCCN  provide an embryo for big droplets at the activation stage  and, more importantly, GCCN enhance droplet growth after activation due to the solute effect."

p. 3, l. 3 sensitivity studies (not sensitivities)

We change "sensitivities" to "sensitivity".

p. 5, l. 24 "kinetic" (i.e., relate to the pace of the process as in chemical kinetics) rather than "kinematic" (i.e., related to motion)?

We change "kinematic" to "kinetic".

p. 6, l. 20 please rephrase "number concentrations of the control"

We rephrase that sentence:

"We test two other aerosol number concentrations , $10^2$ cm$^{-3}$ and $10^4$ cm$^{-3}$, and keep the median radius and geometric standard deviation the same as the control case"

p. 7, l. 15 "larger than **in** the control case"

We add "in" there.

Within references, please correct capitalisation in journal names and use abbreviated versions following the ACP guidelines 2. I strongly suggest adding a doi label for each reference (this will not be added by Copernicus editors). Here are corrections to several entries in the bibliography:

• Bott reference volume should be 59–60.

• Cheng et al. reference is missing page identifier: D08201.

• Falkovich and Pumir reference has wrong year (2015, should be 2007), wrong volume (should be 64) and is missing page numbers: 4497–4505.

• Feingold and Siebert reference is missing book title, editor and publisher information.

• Heintzenberg et al. reference has a truncated title and missing booktitle information, it should likely be replaced with P̈oschl, Rose, and Andreae (2009).

• Laird et al. reference requires correction in capitalisation of "Iii".

• Li et al. reference is missing page range: 11213–11227.

• Lozar and Muessle reference should be cited as "de Lozar and Muessle" (at least according to ACP website).

• Pruppacher and Klett book reference mistakenly includes an additional author and is missing publisher name.

• Xue and Feingold reference is missing page identifier: D18204.

We thank the reviewer for the detailed comments on reference. We corrected our references in the manuscript. DOI number labels for all references are also added.

Figures

It is essential to replace the raster low-resolution image files used in figures 1–6 with vector graphics (PostScript/SVG/PDF formats).

All figures are high-resolution images with *.eps format now.

---

## Author Comment (AC2) · 15 Mar 2018

**Responses to Reviewer 2**

Review of "Cloud droplet size distribution broadening during diffusional growth: ripening amplified by deactivation and reactivation" by Yang et al. submitted for Atmospheric Chemistry and Physics

Using idealized adiabatic parcel simulations with Lagrangian bin-microphysics, the authors investigate the broadening of a cloud droplet size distribution (CDSD). By including the effects of aerosol deactivation and reactivation, it is shown that process of Ostwald ripening, which has been assumed to be weak in warm clouds by other authors, can be significantly amplified, resulting in sufficiently large droplets that might be able to initiate collision and coalescence. The authors demonstrate convincingly that the deactivation of aerosols in a downdraft leads to a lower number of cloud droplets in a subsequent updraft, which enhances the growth of these droplets, resulting in superadiabatic droplets sizes.

Additionally, the reactivation of some aerosols leads to an additional broadening of the CDSD to smaller sizes. Although I feel that the presented results represent a rather extreme case of the amplification of Ostwald ripening due to aerosol deactivation/reactivation, which might not be the case in nature, it clearly demonstrates the effect and potential importance of a proper representation of deactivation/reactivation, which many cloud models lack. Accordingly, some minor additional simulations might be necessary to determine the limits of the presented microphysical processes and to fit it in the current literature. All in all, the manuscript is interesting, well written, and should be published after the following concerns are addressed.

We thank the reviewer for the constructive comments and for the improvements they have motivated in the revised paper. In this document we address all comments and detail the changes in response. Reviewer comments are in blue, our response is in black and modifications of the manuscript are summarized in red text.

General Comments

*Model Description*. Although plenty of references are given, the essential parts of the used microphysical model need to be stated. Only the abstract and the conclusions (Sec. 4) state, that the bin-microphysics is Lagrangian, i.e., it utilizes moving bins instead of fixed bins. This information is missing in Sec. 2 but essential for the model used in this study, which relies on a fixed relation of aerosol mass and droplet size (which is only possible in a Lagrangian (or moving bin) framework). Does the microphysical model include any other processes than diffusional droplet growth including activation/deactivation? Moreover, it would be nice (but not necessary) to present the used equation describing diffusional droplet growth including activation/deactivation. This would be also an opportunity to define quantities as $S_e$ and $S_{sat}$, which are used in other parts of the manuscript (e.g., Fig. 8).

We thank the reviewer for the helpful comments. Section 2 is rewritten with more details and mathematical equations for the employed model. Please see the revised manuscript for more details.

*Idealized Setup.* It is not disputable that the presented simulations represent an idealized setup. However, the probability that a parcel undergoes numerous oscillations of 150 m or more is rather unlikely. The results of Wood et al. (2002), who investigated CDSD ripening in a slightly more realistic setup including potential effects of aerosol deactivation and reactivation (last lines of their section 3), do not indicate a strong evidence of the proposed amplification of CDSD ripening by deactivation/reactivation. Therefore, I strongly suggest testing even thinner recirculation layers, i.e., fluctuations which are more likely to be observed in nature. I expect that if a certain depth of the recirculation layer is undercut, deactivation will be inhibited and the amplification of ripening due to deactivation/reactivation will stop. These additional investigations are not only necessary to understand the importance of the proposed amplification mechanism, but also connects the presented study to other work on spectral ripening (e.g., Wood et al. (2002), or Grabowski and Abade (2017) who extensively investigated the dependence of spectral broadening on the length scales of the involved turbulence in the absence of deactivation/reactivation).

We thank the reviewer for the helpful comments. To investigate whether deactivation or reactivation will be inhibited for thin recirculation layers, three more cases with $\Delta H$=50m, 10m, and 1m are carried out. Results show that reactivation is inhibited, but deactivation always occurs. The evolution of CDSD is similar for those cases, which is similar to the evolution of CDSD due to Ostward ripening in still environment. We add a figure and more discussion in the manuscript.

[Figure]

Figure 7: Cloud droplet size distribution (CDSD) changes with time for different thicknesses of recirculation layers: a) $\Delta H$=50 m, b) $\Delta H$=5 m, c) $\Delta H$=1 m. d) Total cloud droplet number concentration changes with time for the different cases. The gray region in a-c represents the

range of the droplet size spectrum for the control case, and the black lines represent the mean cloud droplet radius change with time.

"We note that deactivation is suppressed for a thin recirculation layer $\Delta H=150$ m as shown in Figure 6b, and therefore the CDSD broadening is not as efficient as the control case. However, the vertical oscillations of an air parcel due to turbulence might be much smaller than 150 m. Wood et al. (2002) did not observe the enhanced CDSD broadening by deactivation and reactivation with a shallower recirculation layer. One interesting question is whether deactivation or reactivation would be inhibited for a very thin recirculation layer. To answer this question, three more cases are carried out with recirculation layers of 50 m, 5 m and 1 m. All these cases have the same setup as the control case except for the thickness of recirculation layer. The CDSD and total cloud droplet number concentration for each case are shown in Figure 7. It can be seen that reactivation is inhibited for all cases, but deactivation always occurs. More interestingly, the CDSD for all these three cases are similar, and the decrease of total cloud droplet number concentration due to deactivation is also similar. The evolution of the CDSD for a thin recirculation layer is independent of air motion and degrades to a steady state where the CDSD broadening is due to Ostward ripening in a still environment."

Minor Comments

P. 2, l. 2: Does the "linear growth rate" refer to the temporal change of the radius ($dr/dt = …$)?

Yes, "linear growth rate" means $dr/dt$. We delete "linear" in our manuscript as suggested by reviewer 1.

P. 3, ll. 30 – 31: Give a short explanation why the liquid water is slightly smaller in the ascending branch compared to the descending.

This is due to the kinetic effect or hysteretic effect that the liquid water content responds slower than the change in environment. In other words, liquid water content is smaller than it should be during condensation (ascent) and is larger than it should be during evaporation (descent). Therefore, the liquid water is slightly smaller in the ascending branch compared to the descending. This is consistent with Korolev et al. (2013). We modify the text as,

"Liquid water mixing ratio in the ascending branch is slightly smaller than that in the descending branch at the same height due to the kinetic effect (or hysteresis effect), which is consistent with Korolev et al. (2013)."

P. 4, ll. 7 – 8: Although it has been stated before, I would mention the development of a second mode in the CDSD by the reactivation of aerosols after about 2 hours.

We modify the text as:

"…Also notice that a second mode appears in the CDSD due to reactivation of aerosols after about 2 hours (see Figure 2d).…"

Fig. 2d: How do the aerosol masses (or dry radii) distribute across the CDSD? I expect that the largest droplets have been grown from the largest aerosols.

Yes, larger cloud droplets include more aerosol mass.

"…Droplet size in the moving bin monotonically increases with the dry aerosol mass associated with that moving bin.…"

P. 5, ll. 34 – 35: Give some more explanations on the setup of the ascending-only parcel simulation. What is its vertical velocity? The answer can be deduced from the following text but a clear statement would be helpful.

We add more details of the ascending-only parcel in the manuscript.

"…For the latter case, the cloud parcel ascends at a vertical velocity of 0.5 ms$^{-1}$ for three hours with the same initial condition as the control case.…"

P. 6, l. 17 and p. 7, l. 12: "Recycling layer"? Based on the available literature, I would prefer the name "recirculation layer".

We replace "recycling layer" and "cycling layer" to "recirculation layer" throughout the manuscript.

P. 6., ll. 27 – 30: Although I agree with the interpretation that deactivation/reactivation might amplify the ripening process, an additional explanation, originating directly from Korolev (1995, Section 2), needs to be considered: Broadening only occurs if the supersaturation is smaller than maximum of S + (r), a quantity which indicates the narrowing or broadening of the spectrum in the vicinity of a certain radius r. If the supersaturations are generally higher than S + , only narrowing of the CDSD occurs. Since in-cloud supersaturations generally decrease due to an increase in aerosol number concentration, it is to expect that only the more aerosol-laden simulations will be affected by Ostwald ripening while the cleaner simulation might be less affected (or not affected at all), which also agrees with the presented study.

We agree with the reviewer's comment. We add another explanation in the manuscript:

"Another explanation from Korolev (1995) is that the CDSD broadening occurs when air supersaturation ($S_e$) is smaller than the critical supersaturation for the smallest cloud droplets ($S_{sat}(r_{small})$). For this condition, the smallest cloud droplets evaporate and the largest cloud droplets might grow slightly if $S_e > S_{sat}(r_{large})$ or evaporate slightly if $S_e < S_{sat}(r_{large})$, thus leading to

broadening. If the water vapor mixing ratio in air on average is much larger than the saturated water vapor mixing ratio over droplet, only narrowing of the CDSD occurs. Because in-cloud supersaturation decreases with increased aerosol concentration, it is expected that the Ostwald ripening is more efficient in polluted cloud, which is also consistent with (Srivastava, 1991)."

Fig. 5b: Where do the high-frequent oscillations in the CDSD come from?

Here we keep the thickness of the recirculation layer constant. Therefore, larger vertical velocity suggests higher oscillation frequency. We add more description in the text:

"…Here we keep the thickness of the recirculation layer constant. Therefore, larger vertical velocity results in a higher oscillation frequency.…"

P. 8, l. 9: For clarity, state the underlying equation used for calculating $S_{sat}$.

We add the equation for $S_{sat}$ in section 2.

P. 8, l. 24 – 34: This is an interesting result. Although I can imagine where the equation in l. 27 comes from, an extra step for its deviations might be illuminating for all readers. Moreover, I suggest discussing the underlying physics of the term sk in slightly more depth. A nice explanation is given for the case of a negative vertical velocity, in which the evaporation of a large number of small droplets maintains the supersaturation at a certain level. But how does sk act in an updraft?

Thank the reviewer for the helpful comments. We add extra step for the deviations in the text.

"This can be obtained from the analytical expression of supersaturation in an adiabatic cloud parcel: $dS_e/dt=Aw-Bnr(S_e-1)$, where A and B are parameters depending on thermodynamic properties (Korolev and Mazin, 2003). …because $dS_e/dt=Aw-Bnr(S_e-S_{sat})$ and thus…"

In the updraft region, all droplets grow and both solute and curvature effects are negligible. We add more discussion in the manuscript:

"In the updraft region, all droplets grow and the effect of $s_k$ is negligible. In the downdraft region and for polydisperse cloud droplets, the environment conditions are buffered by the large number of small cloud droplets."

P. 9, ll. 24 – 25: What is meant by the right upper boundary of the CDSD?

We change "the right upper boundary of the CDSD" to "the large-size upper boundary of the CDSD"

P. 10, ll. 15 – 21: There are models with a similar treatment of microphysics, so-called Lagrangian cloud model. And a couple of publications investigation aerosol activation/deactivation in that framework (e.g., Andrejczuk et al. 2008; Hoffmann et al., 2015; Hoffmann 2017).

We add more discussion about aerosol activation and deactivation in manuscript.

"…The mechanism of CDSD broadening in this study requires the model to consider both solute and curvature effects all the time (i.e., before and after activation, deactivation and reactivation). Our results suggest the importance of solute and curvature effects to the deactivation and reactivation processes, which is consistent with previous studies (e.g., Andrejczuk et al., 2008; Hoffmann et al., 2015; Hoffmann, 2017; Chen et al., 2018)…."

Technical Comments

P. 1, l. 6: Usually, an abstract does not contain any citations.

We removed the citation in the abstract.

P. 1, l. 20: "of a warm cloud" or "of warm clouds"

We modify the text to be "of warm clouds"

P. 2, l. 23: "… GCCN not only **provide** an embryo … but also **enhance** droplet growth …"

We modify the sentence as:

"They found that GCCN  provide an embryo for big droplets at the activation stage  and, more importantly, GCCN enhance droplet growth after activation due to the solute effect."

P. 3, l. 17: Since American English is used throughout the manuscript: "sulfate"

We change "sulphate" to "sulfate" throughout the text.

---

## Author Response (AR2)

**Response to Editor's comments**

We thank the editor for the constructive comments and for the improvements he has motivated in the revised paper. In this document we address all comments and detail the changes in response. Reviewer comments are in blue, our response is in black and modifications of the manuscript are summarized in red text.

The manuscript is an extension of earlier work on oscillating parcels by Korolev (1995) and a variety of papers that have noted Ostwald ripening in cloud parcel models. It does go beyond Korolev (1995) in addressing the relative roles of solute and curvature for drop spectral broadening but it still lacks some perspective, and requires some important, not insignificant changes.

We thank the editor for the helpful comments. We made important changes in the manuscript based on these comments. Especially we added discussion about the updraft-limited and aerosol-limited regimes. We also added a clear and tight conclusion in the end of the discussion.

1) It feels like the study is looking for a niche. Under clean conditions, broadening is weaker whereas in polluted conditions there is progressively more broadening due to deactivation. The base case that is studied is polluted (Na=1000/cm3), presumably because such a case is more appropriate for a model that doesn't include collision-coalescence. The cleaner conditions are relevant for the vast majority of maritime conditions. What importance should be placed on these processes given the idealized nature of the model set-up, especially the absence of collision-coalescence in cleaner conditions and deep clouds?

We agree that the CDSD broadening mechanism due to Ostwald ripening amplified by deactivation and reactivation is more important in polluted conditions, such as continental clouds. For relatively clean conditions such as marine clouds, other CDSD broadening mechanisms might be more relevant, such as collision coalescence or supersaturation fluctuations due to turbulence. We added more discussion on the conditions for which the mechanism might be most important on pages 13-14.

"In summary, the results of this study show that the CDSD can be broadened in a vertically oscillating cloud parcel if both solute and curvature effects are considered, consistent with the findings of previous studies (e.g., Korolev, 1995). Although our model uses an idealized setup, the sensitivity studies help explore the conditions under which this mechanism may be important in the real clouds. The results show that CDSD broadening due to Ostwald ripening can be enhanced in relatively polluted conditions when deactivation and reactivation occur, such as typically exists for continental clouds. For relatively clean conditions like marine clouds, other CDSD broadening mechanisms might be more relevant, such as the collision coalescence process or supersaturation fluctuations due to turbulence."

The study still misses important perspective and has not internalized results from many previous studies. As such it comes across as a bit "pedestrian". Some of the references could be replaced by much older ones. One example is the parcel model work By David Johnson in the late 70s on GCCN. Another important paper that should be confronted: doi:10.1029/2008JD011286 In suggesting further references, I am not asking for a sprinkling of references in a variety of places but rather full engagement with the *ideas* in those papers.

We added those references and more detailed discussion in the manuscript.

"…provides an embryo for large droplets, which can broaden the right branch of the size distribution and can be important for warm rain initiation (Johnson 1982,…)"

"The third condition is that cloud droplets have a long in-cloud residence time, e.g., longer than 1 hour. This is consistent with previous studies that cloud droplet residence time plays an important role in CDSD broadening due to the Ostwald ripening effect (Wood et al., 2002; Romakkaniemi et al., 2009)."

Given the sensitivity of the results to initial aerosol concentrations, a suggestion is to put your results in the perspective of saturation starved conditions vs. conditions where there is ample supply of vapor. Supersaturation starved conditions: Larger nuclei suppressing the activation of smaller particles Ghan et al. (JAS 1998), Feingold and Kreidenweis (2000), Feingold et al. (2001), McFiggans et al. (2006, ACP), Reuter et al. 2009 www.atmos-chem-phys.net/9/7067/2009/

This is a good suggestion. We added a discussion about the two regimes: aerosol-limited regime, where there is ample supply of water vapor, and updraft-limited regime, where supersaturation is starved and larger nuclei will suppress the activation of smaller particles. We also specify our results in each regime.

"In this subsection, we investigate effects of several factors on the CDSD in the adiabatic parcel model with vertical oscillations. Previous studies show that aerosol number concentration and vertical velocity are the two most important factors controlling cloud properties in an adiabatic cloud parcel model (e.g., McFiggans et al., 2006; Reutter et al., 2009; Chen et al., 2018). Two regimes are frequently considered: an aerosol-limited regime exists when there is ample supply of water, and the cloud droplet number concentration is limited by the aerosol number concentration; and an updraft-limited regime exists when supersaturation is starved, and the cloud droplet number concentration is limited by the updraft velocity. In the updraft-limited region, cloud droplets will compete with each other for the limited available water, and the larger aerosols will suppress the activation of smaller aerosols (Ghan et al., 1998; Feingold and Kreidenweis, 2000; Feingold et al., 2001). Based on Reutter et al (2009), the aerosol-limited regime exists when the ratio of the vertical velocity to droplet number concentration, w/n, is larger than $10^{-3}$ m s$^{-1}$ cm$^3$ and the updraft-limited region occurs when the w/n ratio is smaller than $10^{-4}$ m s$^{-1}$ cm$^3$. For the control case, the w/n ratio is $7\times10^{-4}$ ms$^{-1}$ cm$^3$, which is in the transitional regime. In this subsection, we choose several values of aerosol number concentration

and vertical velocity to investigate the CDSD in the aerosol-limited and updraft-limited regimes. In addition, we also test the effect of the recirculation layer thickness on the CDSD broadening. "

"Considering a vertical velocity of 0.5 ms$^{-1}$, they also represent, respectively, the aerosol-limited regime (the $10^2$ cm$^{-3}$ case leads to a w/n ratio of $5\times10^{-3}$ ms$^{-1}$ cm$^3$) and the transition regime (the $10^4$ cm$^{-3}$ case leads to a w/n ratio of $4\times10^{-4}$ ms$^{-1}$ cm$^3$)."

"Results also show that they correspond to the aerosol-limited regime (the 1.0 ms$^{-1}$ case leads to a w/n ratio of $10^{-3}$ ms$^{-1}$ cm$^3$) and the transitional regime (the 0.1 ms$^{-1}$ case leads to a w/n ratio of $5\times10^{-4}$ ms$^{-1}$ cm$^3$)."

Mass accommodation requires more discussion since it can sometimes significantly change drop conc. (e.g., doi:10.1029/2004JD004750 Fig. 10)

We added more discussion about the effect of mass accommodation on drop number concentration and the CDSD in the manuscript.

"In addition, low mass accommodation coefficient inhibits the growth of cloud droplets and leads to more activated cloud droplets (Xue and Feingold, 2004)."

2) Regarding activation/deactivation: Nowhere is it discussed that larger particles may not actually be activated since they have not reached the peak of their Kohler curve. (This doesn't prevent them growing like drops and competing for water vapor.) Please check the particle sizes above which activation doesn't occur and make sure your wording regarding activation/deactivation is accurate. When you claim that particles deactivate, have you checked that they did actually activate? The growth and then subsequent evaporation of droplets as the supersaturation is quenched is well documented. It can be found in text books such as Pruppacher and Klett or Rogers and Yau (see also review by McFiggans et al. 2006, section 3.)

Yes, we carefully checked the critical radius of droplet for each bin. The critical radius of the largest cloud droplet formed on dry aerosol radius of 500 nm is 3.6 µm, and the critical radius for cloud droplets formed on dry aerosol radius smaller than 160 nm is smaller than 1 µm. Figure 2 show that all droplets radii are larger than 4 µm at the end of updraft cycle, suggesting that all cloud droplets are activated at that moment. Because GCCN do not exist in our simulation and the oscillation frequency is low, all cloud droplets have enough time to grow to be activated in the updraft region. We added more discussion in the manuscript.

"It should be mentioned that the critical radius, where the Köhler curve peaks and a droplet is activated, is 3.6 µm for cloud droplet formed on a dry aerosol of 503 nm, and 0.44 µm when formed on dry aerosol of 51 nm. Figure 2d shows that all droplets radii are larger than 4 µm at the end of updraft cycle, indicating that all cloud droplets are activated at that point. Because GCCN do not exist in our simulation and the oscillation frequency is low, all cloud droplets have

enough time to grow to be activated in the updraft region. In this study, we focus on the CDSD at the end of the updraft cycle so the growth and evaporation of unactivated cloud droplets (e.g., McFiggans et al., 2006) will not affect the final CDSD."

3) A parameter that is very relevant but often not plotted/calculated is the drop concentration, n (why is CDNC sometimes used?). It should appear in the Table.

We add drop concentration in Table 1 and change CDNC to n in the manuscript.

4) Pg 2: Line 18: This is incorrect. Broadening can increase or decrease albedo susceptibility depending on the broadening mechanism. (The appropriate papers are quoted but the results misunderstood.)

We correct the statement,

"Previous studies show that an increase in relative dispersion is relevant to the albedo effect and  can either increase or decrease albedo susceptibility depending on the broadening mechanism."

5) Conclusions: Pg 14, line 19: "We have used idealized simulations to find this new CDSD broadening mechanism" is an overstatement given earlier work on this topic. You would do well to highlight the incremental knowledge rather than overstate your case, especially given the highly idealized model set-up. E.g., a 6 km deep cloud will have huge water content and collection/sedimentation processes will be involved. At this point, the history of the drop trajectory through zones of variable qc will be much more important than the fluctuations in S. This ties into discussion on Pg 12 regarding residence time.

We thank the editor for his comments. We modify the text as,

"We have used idealized simulations to  analyze the CDSD broadening in a vertically oscillating cloud parcel due to Ostwald ripening. There are three necessary conditions for this CDSD broadening mechanism.  ... The third condition is that cloud droplets have a long in-cloud residence time, e.g., longer than 1 hour. This is consistent with previous studies that cloud droplet residence time plays an important role in CDSD broadening due to the Ostwald ripening effect (Wood et al., 2002; Romakkaniemi et al., 2009).  We expect that this mechanism of CDSD broadening is possible in the real clouds under those specific conditions."

6) The wording "aerosol particles with different solute effects" implies that you have changed composition across the size distribution whereas my understanding is that what you mean is that larger particles have more dissolved material in them.

We rephrased the sentence based on the editor's comment.

"The first necessary condition is that droplets form on polydisperse aerosol particles  where larger cloud droplets contain more solute."

7) Pg 12: the discussion on the buffered nature of the system vis-à-vis the growth of the largest drops should be juxtaposed with the control that the largest CCN or GCCN have over the system in their ability to control the supersaturation and the activation of smaller drops.

Yes, the system is buffered by smaller CCN when the number concentration of droplets with largest CCN or GCCN is much smaller than that of the droplets with smaller CCN. If all CCNs are larger CCN or GCCN, or in relatively clean conditions, the largest CCN or GCCN will have more effect on the environmental supersaturation. We added more discussion about the effect of larger CCN or GCCN on supersaturation and the activation of smaller drops.

"It should be mentioned that the system is buffered by smaller cloud droplets formed on smaller CCN when the number concentration of those droplets is much more than that for the largest cloud droplets formed on the largest CCN. This might not be true under relatively clean conditions, where the environmental supersaturation can be affected by droplets formed on the largest CCN."

8) The end of the Discussion seems to end without a clear message and could be much tighter.

We added more discussion in the end to make for a clean and tighter conclusion.

"In summary, the results of this study show that the CDSD can be broadened in a vertically oscillating cloud parcel if both solute and curvature effects are considered, consistent with the findings of previous studies (e.g., Korolev, 1995). Although our model uses an idealized setup, the sensitivity studies help explore the conditions under which this mechanism may be important in the real clouds. The results show that CDSD broadening due to Ostwald ripening can be enhanced in relatively polluted conditions when deactivation and reactivation occur, such as typically exists for continental clouds. For relatively clean conditions like marine clouds, other CDSD broadening mechanisms might be more relevant, such as the collision coalescence process or supersaturation fluctuations due to turbulence. When deactivation and reactivation occur, the simulation results show that the smallest cloud droplets do not change significantly after each oscillation cycle, while the largest cloud droplets grow on average after each cycle. The growth of the largest cloud droplet depends on its in-cloud lifetime. This is because, due to the solute effect, the saturation water vapor pressure over larger cloud droplets is smaller than the environmental water vapor pressure that is buffered by numerous smaller cloud droplets with smaller amounts of solute."

Other:

1) While the authors make it clear what the nature of the microphysics scheme is, the term "bin" is not appropriate since it means the number of drops within a bin (i.e., between r and r+dr). The Lagrangian scheme used here is a moving grid scheme not a moving bin scheme. I realise that the reviewers had different thoughts.

Yes, we use a discrete droplet size to represent droplets in a bin. We add more discussion about the scheme used in section 2. We specify that the grid (or bin) used in this study is discrete and not continuous.

"In this study, we use a cloud parcel model with  a moving grid microphysics scheme, where the scheme uses discrete size grids (or bins) to represent droplet sizes."

2) Abstract, line 1: "which determine cloud albedo and rain formation". "lifetime" is a very poorly defined concept in cloud physics.

We change "lifetime" to "rain formation".

3) There are numerous places where the text needs attention vis-à-vis grammar. Please read carefully and correct grammar/tighten the text as necessary.

We thank the editor's comment. We read the text carefully and correct the grammar.

4) Pg 4, line 2: results "confirm"…

We change "show" to "confirm".

5) Pg 5, line 7: remove (no GCCN). It has already been stated.

We remove "(no GCCN)".

6) Pg 7, line 1: "CDSD evolution" is irreversible.

We add "evolution".

7) Pg 8, line 2: rephrase "why and which droplet will deactivate?"

We change it to "Why do some cloud droplets deactivate in the cloud region while others do not?"

8) Pg 8, reference to the quasi steady supersaturation should go back to early papers – e.g., earlier work by Squires (1952), Politovitch and Cooper (JAS, 1988) and later Korolev and Mazin (JAS 2003).

Thank you for the suggestion. We added those references in the manuscript.

9) Comparison with observations (pg 11) should be considered carefully to see if conditions are similar. E.g., the Lu and Seinfeld paper was in a maritime setting with low drop concentrations. You should check the others as well.

We thank the editor for the helpful comment. We corrected the reference and added more discussion in the manuscript,

[revised manuscript text omitted]

---

## Author Response (AR3)

**Response to Editor's comments**

We thank the Co-Editor for the constructive comments. We are glad that he is satisfied with our main changes. In this document we address all comments and detail the changes in response. Reviewer comments are in blue, our response is in black and modifications of the manuscript are summarized in red text.

I am still unhappy with your use of "moving bin" to describe the model framework. The Lagrangian approach uses *discrete sizes* that move, not moving bins. A bin is defined over a range x; x+dx whereas a discrete point is defined at x. I know that the literature is full of usage like "moving bin" but this is incorrect and I would like you to remove all instances and refer to the microphysical approach as a discrete point Lagrangian approach, or similar.

We thank the editor for the helpful comment. We change all "moving bin" to "moving-size grid" and all "bin" to "grid" in the manuscript. To clarify for the reader what we mean, we also modified the description at the beginning of the methods section:

" In this study, we use a cloud parcel model with a moving-size-grid microphysics scheme, where discrete particle sizes on a 1-D grid (initially the radii of dry aerosols [e.g., Table 1]) each grow/shrink according to the environmental conditions to modify the 'moving-size' of the grid element."

Pg 13, line 14/15: "the large number of small ..... buffers the environmental conditions"

We corrected this in the manuscript.

Pg 5, line 2: as an example, of my main point: please state "100 discrete particle sizes". Please address all instances in the manuscript.

We thank the editor's comment. We added a table (now Table 1) in the manuscript to state the dry aerosol radii for all 100 bins.

Pg 14, line 28: the large number...

We corrected this in the manuscript.

[revised manuscript text omitted]